

# Air-Sea fluxes of dimethyl sulphide and methanethiol in the South-West Pacific

Manon Rocco[1,2*], Erin Dunne[3], Alexia Saint-Macary[4,7], Maija Peltola[1], Theresa Barthelmeß[5], Neill Barr[4], Karl Safi[4A], Andrew Marriner[4], Stacy Deppeler[4], James Harnwell[3], Anja Engel[5], Aurélie Colomb[1], Alfonso Saiz-Lopez[6], Mike Harvey[4,†], Cliff S. Law[4,7], and Karine Sellegri[1]

[1] Université Clermont Auvergne, CNRS, Laboratoire de Météorologie Physique (LaMP), 63000, Clermont-Ferrand, France
[2] Now at Instituto de Astronomia, Geofísica e Ciências Atmosféricas/IAG - Universidade de São Paulo - Laboratório de Estudos Ambientais/IPA, São Paulo, Brazil
[3] CSIRO Ocean & Atmosphere, Aspendale, Australia
[4] National Institute of Water and Atmospheric Research, Wellington, New Zealand
[4A] National Institute of Water and Atmospheric Research, Hamilton, New Zealand
[5] GEOMAR Helmholtz Centre for Ocean Research Kiel, Kiel, Germany
[6] National Center for Atmospheric Research, NCAR, Boulder, USA
[7] Department of Marine Science, University of Otago, New Zealand
[†] Deceased

*Correspondence to*: Karine Sellegri (karine.sellegri@uca.fr); Manon Rocco (rocco.manon@gmail.com)

**Abstract.**

Air-sea fluxes of dimethyl sulphide (DMS) and methanethiol (MeSH) from surface seawater in the remote Southern Pacific Ocean were measured in three Air-Sea Interface Tank (ASIT) experiments during the Sea2Cloud voyage in March 2020. The measured fluxes of $0.78 \pm 0.44$ ng m$^{-2}$ s$^{-1}$ and $0.05 \pm 0.03$ ng m$^{-2}$ s$^{-1}$ for DMS and MeSH, respectively, varied between experiments reflecting the different water mass types investigated, with lowest fluxes with subtropical water and highest with biologically-active water with sub-Tropical water and highest from the sub-Tropical Front. Measured DMS fluxes were consistent with calculated fluxes from a two-layer model using DMS concentration in the ASIT seawater. The experiments also determined the influence of elevated ozone, with one ASIT headspace amended with 10 ppbv ozone while the other provided an unamended control. Elevated ozone resulted in a decrease in DMS flux, corresponding to decreased conversion of dimethylsulfoniopropionate (DMSP) to DMS in the seawater. The MeSH:DMS flux range was 11-18% across experiments, in line with previous observations, indicating that MeSH represents a significant contribution to the atmospheric sulfur budget. Using the ASIT results in combination with ambient seawater concentrations during Sea2Cloud, significant linear correlations were identified for both DMS and MeSH fluxes with nanophytoplankton cell abundance ($r_{DMS}= 0.73$ and $r_{MeSH}= 0.86$), indicating an important role for this phytoplankton size class, and also its potential as a proxy for estimating DMS and MeSH emissions in chemistry-climate models.

## 1 Introduction

Oceanic emission of dimethyl sulphide (DMS) is considered the largest natural source of atmospheric sulfur, with the global flux estimated at 23 - 35 Tg S yr$^{-1}$ (Simó and Dachs, 2002; Lana et al., 2011). In the remote marine atmosphere, the oxidation of DMS leads to the formation of sulfuric acid, a key species for aerosol nucleation,



and is considered to be a major source of cloud condensation nuclei (CCN) in the remote marine atmosphere (Korhonen et al., 2008). The CLAW hypothesis (Charlson et al., 1987) proposed that enhanced DMS emissions would lead to higher numbers of CCN and so increased cloud albedo, subsequently cooling Earth's temperature. The proposition of the CLAW hypothesis has been the stimulus for intensive research on the cycle of DMS and its role in atmospheric chemistry and climate processes (Kloster et al 2006).


In the surface ocean, DMS is produced from the degradation of dimethylsulfoniopropionate (DMSP), which is produced by marine macroalgae, phytoplankton or bacteria (Bentley and Chasteen, 2004; Kloster et al., 2006; Novak and Bertram, 2020). To form DMS, the DMSP undergoes reactions catalysed by DMSP lyase, (Taylor and Visscher, 1996; Steinke et al., 1996; Kiene, 1996a) and non-enzymatic pathways of demethylation (Bentley and

Chasteen, 2004). Furthermore, DMS can be produced by the biological reduction and oxidation of DMSP and also abiotically by light-dependent reactions (McNabb and Tortell, 2021 and references therein). These multiple chemical and biological pathways of DMS flux make numerical prediction of DMS complex. Another product of DMSP degradation is methanethiol (MeSH) (Kiene, 1996b) for which production is equivalent to ~17% of DMS from marine sources (Lee and Brimblecombe, 2016). MeSH has a shorter lifetime than DMS in the atmosphere,

of a few hours relative to ~1 day (Lee and Brimblecombe, 2016). However, the atmospheric fate of sulfur emitted as MeSH, including its role in marine aerosol formation, is highly uncertain due to the scarcity of data on emissions, atmospheric concentrations and photochemical processing of this species. Indeed, the most recent studies by Lawson et al., (2020) and Kilgour et al., (2021), concluded that MeSH fluxes are underestimated by a factor of 4 in earlier studies by Kettle et al. (2001).


DMS is the main (often only) marine compound of biogenic origin implemented in regional (Marelle et al., 2017) and global (Carslaw et al., 2013; Woodhouse et al., 2013; Mahajan et al., 2015) atmospheric models. However, the role of DMS in climate regulation has been questioned (Quinn and Bates, 2011), and there is no consensus on the net effect of climate change on DMS emissions within the modelling community. Most studies predict a future

increase in global DMS concentrations (Bopp et al., 2003), while other models predict a decrease under future scenarios (Kloster et al., 2007; Schwinger et al., 2017), with some divergence between predictions arising from DMS concentration input to the models. For deriving a sea-to-air DMS flux, several modelling exercises use DMS concentration in the seawater derived from the Lana et al. (2011) climatology (Mahajan et al., 2015; Marelle et al. 2017), while others use the output from ocean biogeochemical models (Elliott et al. 2009). There are also

discrepancies in the way DMS concentrations in the seawater are prescribed in ocean biogeochemical models. For example, DMS production is represented in ocean biogeochemical numerical models by simulating the relationships between different phytoplankton classes with zooplankton, light, temperature and nutrient availability (Vogt et al., 2010), each of which include some degree of uncertainty.

As ocean biogeochemical models are complex tools, there have also been recent attempts to generate DMS fields from a simplified set of biogeochemical parameters. In Wang et al., (2020), a large database of DMS measurements in surface waters was used with environmental parameters (latitude-longitude, time of day and year, solar radiation, mixed layer depth, sea surface temperature and salinity, chlorophyll-a (Chl-a) and nutrient availability) to predict oceanic DMS in a neural network approach. The resulting multi-linear regression between



DMS concentration and these parameters only captured around 30% of the variance in DMS, and strongly underestimated DMS in regions of high concentrations. Galí et al., (2018), have estimated DMSP from satellite-retrieved Chl-a and light and then sea-surface DMS as a function of DMSP and photosynthetically available radiation (PAR). More recently, Bell et al., (2021) used a set of DMS measurements from four ship campaigns in the North Atlantic to compare the Lana et al. (2011) climatology with the Galí et al., (2018) and Wang et al.,

(2020) approaches. These authors observed that the Lana et al (2011) DMS climatology provided an adequate seasonal variation of the DMS concentrations but failed to capture variability in DMS over short spatio-temporal scales, while the Galí et al. (2018) algorithm and neural network model outputs under-predicted measured DMS concentrations in specific areas such as the Southern Ocean. Gali et al., (2018) conclude that these recent algorithms and models may be limited because their input variables do not encapsulate all of the key biological

processes involved. In particular, Chl-a may not be an adequate biological variable to predict DMS concentrations, as indicated by field studies and modelling, as it is present in all phytoplankton groups whereas DMS production varies with taxa. Laboratory and mesocosm experiments have shown that DMS is preferentially associated with two phytoplankton groups: dinoflagellates and coccolithophores (Yassaa et al., 2006; Kwint and Kramer, 1995; Kwint et al., 1993; Levasseur et al., 1996), consistent with the elevated cellular DMSP content of these groups

(Keller and Korjeff-Bellows, 1996). Furthermore, environmental factors such as light stress and temperature may also modulate DMS and MeSH production by phytoplankton species (Kameyama et al., 2011).

In the present paper, we describe the use of an original experimental set-up with the application of Air-Sea Interface Tanks (ASIT), to measure DMS and MeSH sea-air fluxes in three different seawater types collected

during a voyage in the South-West Pacific. Due to atmospheric mixing and transport, the atmospheric concentrations of DMS, MeSH and other trace gases in ambient air are spatially dislocated from their marine sources and sinks (Bell et al, 2015), limiting our capacity to directly investigate air-sea interactions via underway observations of ambient air and seawater. While less representative of open ocean conditions in term of wind-derived processes, the novel ASIT experiments employed in the Sea2Cloud voyage allowed direct observation of

the air-sea net fluxes of DMS and MeSH, their relative contribution to the marine sulfur emission budget, and their link with underlying biogeochemistry of distinct seawater types, without the confounding influences from air- and waterside turbulence, precipitation, and bubble-bursting. This experimental set-up also provided the capacity to control some variables. As surface ozone has increased in clean Southern Hemisphere air over the last 30 years and is projected to continue to rise (Cooper et al. 2020), we evaluated the impact of ozone-mediated

oxidative stress on these fluxes. In addition, the relationship between fluxes and seawater concentrations with physical, chemical and biological seawater properties were explored to develop DMS flux parameterizations for potential application in modelling efforts to constrain the spatial and temporal distribution of DMS fluxes in this region.

## 2 Materials and Methods

The experiments were conducted onboard the *R/V Tangaroa* during the Sea2Cloud voyage in the South-West Pacific Ocean east of New Zealand, around the Chatham Rise (44ºS, 174–181ºE), in the late austral summer from 17 to 27 March 2020 (Sellegri et al., 2022). The Chatham Rise represents the junction where sub-Antarctic seawaters meets the sub-Tropical seawater and supports blooms of high phytoplankton abundance and diversity



along the sub-Tropical Front (Law et al., 2017). The Sea2Cloud voyage objectives and measurements are
summarised in (Sellegri et al., 2023).

**2.1 Air-Sea Interaction Tanks (ASITs)**

During the Sea2Cloud voyage, two Air-Sea Interface Tanks (ASITs) were deployed for semi-controlled studies
of sea-air exchange from seawater of differing origin: sub-Antarctic; sub-Tropical and Frontal. The ASITs
consisted of two cylindrical chambers, each of 1.82 m$^3$ volume, lined with Teflon film and enclosed by a
transparent lid composed of PMMA to minimise loss of short-wave radiation. The water temperature in the ASITs
was maintained at ambient surface temperature by a heat exchanger, with water and headspace temperature and
light conditions continuously monitored in both tanks (Ecotriplet; HOBO Pendant temp/light, Onset, Bourne MA
USA). The ASITs were mounted on the rear deck of the vessel with in-built baffles in each to reduce turbulence
and mixing arising from movement of the ship. Ambient air was drawn from above the bridge of the ship via a
400 mm ECOLO Polyurethane Antistatic hose at 1000 L min$^{-1}$, a subsample of which was pumped (Gast
Manufacturing, MI, USA) via a particle filter through the headspace of each ASIT at ~23 L min$^{-1}$, resulting in a
residence time of ~ 40 minutes. One ASIT had an additional 10 ppbv of ozone continuously added to the headspace
(ASIT-O$_3$) using an ozone generator (MGC101, Environmental S.A., Poissy, France), while the other was not
amended and provided a control (ASIT-control). The ASIT-O$_3$ ozone concentrations (14.5 ± 2.9 ppbv) were closer
to the ambient air ozone concentrations (14.6 ± 1.8 ppbv) than the ASIT-control levels (6.7 ± 1.5 ppbv) due to
wall losses of ozone in the tanks and sampling lines.

An air-conditioned shipping-container laboratory was located on the rear deck adjacent to the ASITs which housed
a suite of gas and particle monitoring instruments connected to a common sampling manifold (Figure 1). The air
being sampled was controlled by a 4-way electronic valve (TSI) that switched every 20 min to sequentially sample:
(1) ambient air via an ~5 m stainless steel inlet (OD 100mm) with the outlet located over the port side of the ship
and a flowrate of ~20 L min$^{-1}$; (2) ASIT-control, and (3) ASIT-O$_3$, each via ~ 2.14 m of ⅜ inch stainless steel
inlets; and (4) the headspace flush air prior to entering the ASITs (ASIT bypass, See Figure 1).

There were variations in the irradiance between the two tanks due to reliance upon natural light on deck.
Continuous monitoring of incident PAR showed that cumulative irradiance differed by less than 20% between the
two ASITs at the end of the 48-hour experiments for EXP B and C, but was about 40% higher in the ASIT-control
compared to ASIT-O$_3$ during EXP A on the 22nd of March.

**2.2 Seawater sampling and analysis**

At the start of each experiment, the two ASITs were flushed with ambient seawater collected from 3-m depth
using a towed fish deployed 3 m from the side whilst the ship was in motion to avoid contamination. The flushing
via previously acid-washed piping lasted for 3 hours, after which the two ASITs were filled simultaneously to a
volume ~ 0.9 m$^3$ and then sealed with ~1 m$^3$ of headspace air overlying the seawater. Three distinct sea water
types were used in the individual ASIT experiments (Table 1, Figure 2). Sub-Tropical frontal waters were
incubated in EXP A for the period 20/03 06:00 LT – 22/03 23:00 LT; sub-Antarctic water were incubated in EXP



B for the period 23/03 06:00 LT – 25/03 06:00 LT; and sub-Tropical water incubated in EXP C for the period 25/03 15:00 LT – 27/03 06:00 LT.

Seawater samples were collected from each ASIT at the start of each experiment at 06:00-09:00, and repeated each day of the experiments, via a gravity-fed outlet pipe, with a total of seven seawater aliquots collected from each ASIT over the course of the three experiments. The daily seawater samples were analysed for DMSw and DMSP concentration (see Saint-Macary et al 2022 for method), with replicate samples collected on day 0 of EXP A agreeing within 6%. Other seawater biogeochemical parameters analysed included Chlorophyll-a (Chl-a), dissolved and particulate nitrogen (DN, PN) and particulate carbon (PC), chromophoric dissolved organic matter (CDOM), dissolved organic carbon (DOC), dissolved hydrolysable amino acids (DAA) and combined carbohydrates (DCHO), and also microbial community composition (flow cytometry, flowcam and microscopy). The analysis of the dissolved organic matter components excluded all particles larger than 0.45 µm. Further details of these seawater measurements are provided in Sellegri et al. (2023) and in the supplement of this paper. Following each 2-day experiment the ASITs tanks were drained and cleaned.

## 2.3 Analysis of ASITs headspace and ambient air

A Proton Transfer Reaction – Quadrupole Mass Spectrometer (PTR-MS, Ionicon Analytik, Innsbruck, Austria) was used to measure VOCs in the ASIT headspace and ambient air. This technique has been described in detail elsewhere (Blake et al., 2009; Gouw et al., 2003; Lindinger et al., 1998). Briefly, the PTR-MS was operated with an inlet temperature of 60 °C, an applied voltage of 600 V and pressure of ~2.0 mbar in the drift tube reaction chamber. The ion source produced primary reagent ion signals $H_3O^+$ with a purity of ~97%. The PTR-MS was operated in multiple ion detection mode and scanned 24 selected masses with a 10 s second dwell time. The instrument produced a mass scan every ~3 minutes. In PTR-MS measurements of the marine atmosphere the ion signals at m/z 49 and m/z 63 are typically attributed to the parent ions of MeSH and DMS (Lawson et al 2020). Production of other compounds can also contribute to these m/z (Kilgour et al., 2021), and cannot be resolved with the PTR-MS employed in this study, so reported concentrations of DMS and MeSH should be considered an upper estimate.

Zero-air measurements were performed daily by-passing ambient air through a platinum wool catalyst heated to 400 °C in order to remove VOCs while maintaining the same mole fractions of the other natural components of air ($N_2$, $O_2$, $CO_2$, $H_2O$ etc). Zero measurements were performed for each ASITs, averaged and subtracted from the measured ASIT and ambient VOC sample concentrations. Calibrations were performed with certified gaseous standards (Apel Riemer Env, Inc, Broomfield, CO) containing a mixture of VOCs including ~1 ppm DMS in nitrogen (Stated accuracy ± 5%). The PTR-MS sensitivity to DMS was 8.96 ncps/ppbv in ASIT-control and 8.36 ncps/ppbv in ASIT-$O_3$ and 4.43 ncps/ppbv empirically derived calibration factor for MeSH in both ASITs, respectively. This is consistent with recent PTR-ToF studies of DMS and MeSH which have reported sensitivities to DMS and MeSH of 3.9 cps/ppt and 1.3 cps/ppt, respectively in Novak et al., (2022) and 3.0 cps/ppt and 1.0 cps/ppt, respectively in Kilgour et al., (2021).

The minimum detection limit (MDL) for a single measurement was determined from the scatter in the zero measurements and set at the 95th percentile of the deviations about the mean zero. 100% of the DMS observations



collected in the ASIT were greater than the detection limits (MDL (ASIT-control) = 78 pptv, MDL (ASIT-$O_3$) = 107 pptv). 40% of the observations of MeSH in the ASIT-control were greater than the MDL (15 pptv) and 65% of the observations in the ASIT-$O_3$ were greater than the MDL (22 pptv). In the ASITs experiments the concentration of DMS and MeSH measured in the bypass air was subtracted from the concentrations measured in the ASITs air in order to quantify the enhancement in DMS and MeSH due to emissions from seawater in the

ASITs.

Ozone and sulfur dioxide ($SO_2$) were continuously measured with a UV photometric analyzer (TEI49i, Thermo Fisher Scientific, Waltham, MA USA) and a $SO_2$ analyzer using pulsed fluorescence (TEI43i, Thermo Fisher Scientific, Waltham, MA USA). Meteorological parameters experienced during the voyage were measured by an

automatic weather station (AWS) mounted on top of the crow's nest above the bridge.

Contamination of the flushing air from the ship exhaust was filtered out using a $SO_2$ threshold of 0.2 ppbv; when a spike of $SO_2$ was observed in the bypass air, data in the ASITs were filtered out from this time on for the following 2 h 15 min, based on the fact that, at the flow rate of 25 L min$^{-1}$ 90% of the ASITs headspace air is changed in 2 h 15 min. At least this time-lag was applied after the lid of the ASITs was closed at the beginning of

each experiment, for which fluxes measurements were discarded, ensuring that the headspace was fully flushed and reached equilibrium with the seawater below.

**2.4. Equilibrium in the ASITs**

For a given seawater concentration of DMS, the concentration of DMS in the headspace expected at thermodynamic equilibrium can be estimated using the Henry's law equation:

$$VOC_{air} = \frac{VOC_{water}}{(f_{sal} * H_{(T)})} \qquad \text{(Eq. 1)}$$

where $f_{sal}$ is the factor that accounts for the salinity of seawater ($f_{sal}$ = 4.5345), calculated from the concentration-based Sechenov coefficient ks and the concentration of salt (mol/kg), and H (T)s calculated using the Henry law constant (Sander, 2014):

$$H(T) = H_0 exp\left(4300 \times \left(\frac{1}{T} - \frac{1}{T_0}\right)\right) \qquad \text{(Eq. 2)}$$

where $H_0$= 4.2 x 10$^{-3}$ mol/(m$^3$ Pa) for DMS and $H_0$ = 3.8 x 10$^{-3}$ mol/(m$^3$ Pa) for MeSHw , T is ambient temperature

and $T_0$ is reference temperature of 298.15 K. Relatively constant temperatures were maintained during each experiment and would of had minor influence on Henry's law partitioning between aqueous and the gas phase (Sinha et al 2007, Rocco et al 2021b).

**2.5 Determination of net sea-air fluxes of DMS and MeSH from ASITs headspace concentrations**

Assuming equilibrium conditions were established in ASITs, the net fluxes of DMS and MeSH can be determined

from the headspace concentrations measured by the PTR-MS, the ASITs geometry and headspace air flow rates, as described in Equation 3, and previously described by Sinha et al., (2007) and previous reports from this study (Rocco et al., 2021).



$$F_{VOC} = \frac{Q}{A} \times \Delta[X]_{ASITs}(ppb) \times \frac{M_{VOC}}{V_m}$$  (Eq. 3)

$F_{VOC}$ is the flux of VOCs in the ASITs in µg m$^{-2}$ s$^{-1}$, Q is the flow rate of the bypass air into the mesocosm, A is the surface area of the seawater enclosed in the ASITs in m$^2$, $M_{VOC}$ is the molecular weight of X compound in g mol$^{-1}$, $V_m$ is the molar gas volume in m$^3$ kmol$^{-1}$ (23.233 at 1015.25 hPa and 283 K) and $\Delta[X]_{ASITS}$ (ppbv) = $[X]_{ASITS}$ (ppbv) - $[X]_{bypass}$ (ppbv), where $[X]_{ASITS}$ (ppbv) is the concentration in the ASITs and $[X]_{bypass}$ (ppbv) the concentration in the bypass air. The ratios of $[X]_{ASIT}/[X]_{bypass}$ ranged from 2.3 – 5.2 across the three experiments and variability in the reported fluxes was not dominated by variability in the composition of the bypass air.

**2.6 Determination of sea-air fluxes of DMS from ASITs seawater concentrations**

Regional and global models often utilise fluxes of DMS determined from DMS concentrations in the seawater (e.g. Marelle et al., 2016, Bopp et al., 2003, Lana et al 2011). Using the measured DMS concentrations in the ASIT seawater, we calculated the DMS fluxes $F_c$ (g cm$^{-1}$ h$^{-1}$) to the atmosphere following the approach described by Saltzman et al., (1993):

$$Fc = k_{flx}(-C_g + \alpha C_l)$$  (Eq. 4)

where $k_{flx}$ is the gas exchange coefficient, $C_g$ and $C_l$, the concentrations in the gas and liquid phase, respectively and α is the dimensionless solubility of the gas in the seawater, which is expressed by the McGillis et al., (2000) equation in Eq 6 and $k_{flx}$ by Eq 7. from Wanninkhof, (2014):

$$\alpha = e^{\left(\frac{3525}{SST} - 9,464\right)}$$  (Eq. 5)

$$k_{flx} = 0,251 \left(\frac{660}{S_c(DMS)}\right)^{\frac{1}{2}} u^2$$  (Eq. 6)

where Sc is the Schmidt number defined by Saltzman et al., (1993):

$$S_{c(dms)} = 2674 - 147,12 \times SST + 3,726 \times SST^2 - 0,038 \times SST^3$$  (Eq. 7)

in which SST is sea surface temperature in °C and $u$ is wind speed, usually normalised to the height of 10 m above the ocean surface in ambient conditions. We calculated DMS fluxes from the seawater DMS concentrations using the set of equations used in regional modelling (section 2.7.), i.e. including the wind effect (therefore adding kinetics compared to the Henry's law equilibrium used to plot Figure 3). Calculated DMS fluxes ($F_c$) fitted measured fluxes ($F_{VOC}$ in both ASITs, Figure 3) with a slope close to 1 in the ASIT-control for an equivalent wind speed in both ASITs was 0.59 m s$^{-1}$.

These results indicate that the fluxes measured in the ASIT systems are equivalent to those that would be modelled from DMSw for an equivalent wind speed of 0.59 m s$^{-1}$ in ambient air.

**2.7 Determination of DMS and MeSH fluxes from ambient MBL concentrations**

The ambient fluxes of DMS and MeSH in the marine boundary layer (MBL) was calculated using the equation described by Marandino et al., (2009) and applied by Lawson et al (2020) and Rocco et al. (2021):

$$F_{ambientDMS} = \frac{d[C]}{dt} \times h_{MBL}$$  (Eq. 8)



Here, C is the concentration in ng m$^{-3}$, dt is the difference of time between the measurement of the highest and the lowest concentration of DMS and h$_{MBL}$ the nocturnal Mixed Boundary layer (MBL) in metres determined by radiosonde measurements (range between 670 m and 1450 m for the whole campaign). F$_{ambient\ DMS}$ is the flux of DMS in ambient air in ng m$^{-2}$ s$^{-1}$ deduced from nocturnal DMS measurements. This flux is estimated based on the assumption of minimal oxidation of DMS during the nighttime which favours the nocturnal accumulation of primary DMS. The highest levels of DMS concentrations were observed at ~06:00 LT and the lowest at ~17:00 LT. Three nights without terrestrial influence were selected for the calculation: from 21 March 21:00 LT to 22 March 06:00 LT, from 22 March 20:00 LT to 23 March 00:00 LT and from 23 March 20:00 LT to 06:00 LT, with corresponding MBL heights of 1200 m, 670 m and 770 m, respectively.

## 3. Results and discussion

### 3.1 Mixing ratios and fluxes of DMS and MeSH in the marine boundary layer

Atmospheric mixing ratios of dimethylsulfide (DMSa) and methanethiol (MeSHa) sampled via the ambient inlet over the voyage track are shown in Figure 4. Mixing ratios of DMS ranged from below the detection limit (< 78 pptv) to 1285 pptv with a voyage average of 185 ± 184 pptv, whereas MeSH ranged from below detection limit (< 15 pptv) to 150 pptv with a voyage average of 40 ± 28 pptv. The highest concentrations of DMS and MeSH were observed over the frontal bloom waters (Figure 4) and were similar to those from a previous voyage over phytoplankton blooms in this region with a reported DMS average of 208 ppt, ranging up to 987 ppt and maximum of ~1000ppt (Bell et al, 2015), and a MeSH average of 18 ppt, ranging up to 65 ppt (Lawson et al 2020).

Likewise, when both species were detectable the relationship between MeSH and DMS yielded a slope of 0.13 (R$^2$ = 0.52), which was almost identical to the relationship previously reported by Lawson et al over a coccolithophore bloom in this region (Figure 5, Slope = 0.13, R$^2$ = 0.5, Lawson, et al 2020). The correlations between DMS and MeSH are reflective of a common seawater source.

Fluxes were determined from night-time marine boundary layer (MBL) concentrations of DMS and MeSH using the nocturnal accumulation method (Sect 2.7). Flux results for each of the three nights separately average 15.39 ng m$^{-2}$ s$^{-1}$, 38.33 ng m$^{-2}$ s$^{-1}$ and 7.75 ng m$^{-2}$ s$^{-1}$ for DMS and 2.43 ng m$^{-2}$ s$^{-1}$, 2.98 ng m$^{-2}$ s$^{-1}$, 0.26 ng m$^{-2}$ s$^{-1}$ for MeSH. These values are in line with the values measured in Lawson et al. (2020) with fluxes ranging between 9.1 - 22.3 ng m$^{-2}$ s$^{-1}$ for DMS and 1.9 - 3.2 ng m$^{-2}$ s$^{-1}$ for MeSH. The ratio of F(MeSH)/F(DMS) is between 0.03 and 0.15, similar to that of Lawson et al. (2020) who obtained a ratio of 0.15 for a mixed community bloom of coccolithophores, flagellates and dinoflagellates. In our study, this ratio was measured for the first night (21-22 March) when the ship was located over a mixed bloom of diatoms, *Synechococcus* and dinoflagellates. Overall, these studies confirm the biologically productive subtropical front as a hotspot for sulphur emissions (Lizotte et al, 2017), for both DMS and MeSH. Given the higher reactivity of MeSH with OH, this source could be important for SOA formation and atmospheric oxidation capacity in this region.

### 3.2 DMS and DMSP in the ASITs headspace and seawater

During the experiments the mixing ratios of DMS in the ASIT-control headspace were on average > 10 times the DMS in the incoming flushing air and 6 times the flushing air in the ASIT-O$_3$ headspace, indicating that DMS



emission from the seawater typically exceeded loss processes in the headspace (e.g. deposition, chemical transformation, wall effects). The estimated headspace concentration of DMS calculated from the Henry's law equation (Eq. 1) showed good agreement with measured DMSa (slope =1.66, intercept = -0.75, $R^2$ = 0.94), indicating equilibrium conditions were established in the ASITs (Figure 6).

The lifetime of DMS relative to OH oxidation is estimated to be between 16 hours to 1 day (Novak et al., 2022, Lee and Brimblecombe, 2016; Lawson et al., 2020), and the lifetime of DMS due to ozone is 15 days (Vrekoussis et al., 2004). The correlations between DMS in air and seawater did not differ between the ASIT-control and ASIT-O$_3$ (slope = 0.51 ± 0.03; intercept = -0.68 ± 0.16; $R^2$ = 0.97; slope = 0.33 ± 0.04; intercept = -0.32 ± 0.16; $R^2$ = 0.88, respectively, Figure S.1) indicating that the short residence time (~ 40 min) limited the contribution of chemical losses to the observed concentrations in the ASITs headspace, and changes in headspace concentrations reflected changes in dissolved DMS concentrations in the underlying seawater.

The highest DMS headspace concentrations were observed with frontal seawater during EXP A, with peaks up to ~ 6 ppbv in the ASIT-control, and ~ 2.5 ppbv in the ASIT-O$_3$ (Figure 7 and Figure S.2). Moderate headspace mixing ratios up to 2 ppbv in the ASIT-control and ~1 ppbv in the ASIT-O$_3$ were observed during EXP B with sub-Antarctic water, whereas lowest headspace mixing ratios were obtained with sub-Tropical water in EXP C with both ASIT-control and ASIT-O$_3$ showing values of ~ 0.5 – 1 ppbv. In turn, the concentrations of DMSw were closely related to the concentration of the precursor DMSP ($R^2$ = 0.65, Figure S. 3 and Figure S.4), with the exception of an outlier DMSP and DMS data point in the ASIT-control at the end of EXP A (Figure 7b).

Over the course of EXP A the levels of DMSw in the ASIT-control doubled from 5 to 11 nM accompanied by a decrease in DMSP from 94 to 67 nM (Figure S.5). Conversely, only small changes occurred in DMSw (from 4 to 6 nM) and DMSP (from 100 to 93 nM) in the ASIT-O$_3$ during EXP A. These were the largest changes in DMSw and DMSP, as differences between the ASIT-control and ASIT-O$_3$ were less pronounced in EXP B and C, at < 2nM in DMSw and < 5 nM in DMSP between the two treatments (Fig 5). The higher values in frontal waters was consistent with concomitant deckboard incubations of sea surface microlayer water, which also showed highest DMSP loss and DMS production highest in frontal waters at the same location (Saint-Macary et al 2022).

The concentration of DMS in the ASITs were also consistent with concentrations measured in ambient surface seawaters of 2.41 ± 1.59 nM, 1.98 ± 0.32 nM and 1.52 ± 0.34 nM in EXP A, B and C, respectively and also in workboat samples (WKB) collected at distance from the vessel, of 1.38 ± 0.19 nM, 2.33 ± 1.24 nM and 1.46 ± 0.16 nM. Moreover, the higher dissolved DMS concentrations in the frontal seawater were representative of levels previously reported from frontal blooms in this region (e.g. SOAP Voyage, 8.1 nM) (Walker et al 2016), whereas the dissolved DMS levels in the sub-Antarctic water and sub-Tropical water samples were similar to those reported in previous studies at sub-Antarctic and sub-Tropical latitudes (Dani et al., 2017), indicating the ASITs seawater samples were broadly representative of water masses in these regions.

The DMSP method used in this study determined total DMSP which is composed of both dissolved and particulate (intracellular) DMSP. Approximately 80% of DMSP is in the particulate form within phytoplankton cells, and not available for bacterial catabolism to DMSw (Keller and Korjeff-Bellows, 1996; Belviso et al., 2000; Yang et al.,





2005a; Zhang et al., 2009). Dissolved DMSP released from cells via grazing, viral lysis and cell lysis during senescence provides the source for bacterial catabolism to DMSw. The changes in DMSw and DMSP observed in EXP A could be interpreted as indicating that a greater proportion of DMSP was available in dissolved form

for conversion to DMSw due to greater grazing, viral lysis or phytoplankton senescence in the ASIT-control. Alternatively, the data may suggest that the enzymatic cleavage process that converts DMSP to DMSw was more active in the ASIT-control than in the ASIT-O₃. This could involve either enzymatic cleavage of dissolved DMSP by bacteria, and/or via direct cleavage of intracellular DMSP to DMS within certain phytoplankton types (Lizotte et al., 2017). Given the proposed anti-oxidant functions of DMS and DMSP (Sunda et al., 2002) it would be

reasonable to expect increased DMSP and DMS production in ASIT-O₃, although in the ASIT-O₃ headspace O₃ levels were typical of ambient air (~15 ppbv).

**3.3 MeSH and its relationship to DMS and DMSP**

Given similarities in their Henry's Law and diffusion constants, the magnitude of the sea-air fluxes of DMS and

MeSH is likely driven by competing waterside biological production and loss processes, which were further explored in the ASITs. Conversion to DMS is only a minor (5 - 10%) pathway for DMSP removal, which is instead predominantly removed via bacterial demethylation or demethiolation of DMSP to MeSH (Kiene and Linn, 2000). Larger mixing ratios of DMS than MeSH can be explained rapid conversion of MeSH to bacterial protein sulfur (Keine et al. 1999). Figure 8a shows the relationship between DMSa and MeSHa headspace

concentrations in each of the ASITs. As for DMS, MeSH atmospheric lifetime (9 h) is many times greater than the residence time in the ASITs headspace (~ 40 min) and therefore we assumed chemical loss processes did not significantly influence MeSHa within and between experiments and between the two different ASITs.

The highest MeSH headspace concentrations were observed with the frontal seawater in EXP A with peaks up to

~ 0.9 ppbv in the ASIT-control (average = 0.4 ± 0.2 ppbv), and up to ~ 1 ppbv in the ASIT-O₃ (average = 0.5 ± 0.2 ppbv) (Figure 8). Moderate headspace mixing ratios up to 0.5 ppbv in the ASIT-control (average 0.2 ± 0.1 ppbv) and peaks up to 0.4 ppbv in the ASIT-O₃ (average = 0.1 ± 0.1 ppbv) were observed during EXP B with sub-Antarctic water, with lowest headspace mixing ratios from the sub-Tropical water in EXP C with ASIT-control average of 0.09 ± 0.06 ppbv and ASIT-O₃ average values 0.02 ± 0.05 ppbv.


Although MeSH concentration was not determined in the ASITs seawater, the similarity in Henry's Law constant with DMS supported the assumption of equilibrium conditions for MeSH Dissolved MeSH (MeSHw) concentrations were calculated from the MeSHa assuming thermodynamic equilibrium according to the Henry's law (Eq 1) and are shown in Figure S.6 and Figure S.7. Calculated MeSHw concentrations for frontal seawater

(EXP A) was 1.02 ± 0.62 nM in the ASIT-control and 1.66 ± 1.34 nM in the ASIT-O₃. For the sub-Antarctic seawater samples (EXP B) MeSHw concentrations of 0.74 ± 0.25 nM for the ASIT-control and 0.32 ± 0.16 nM for the ASIT-O₃. For the sub-Tropical seawater samples (EXP C) the ASIT-control MeSHw concentrations were 0.31 ± 0.08 nM, and in the ASIT-O₃ between 0.11 ± 0.07 nM. These estimated MeSHw concentrations are within the range of concentrations reported in previous studies of ~ 0.2 nM in the Baltic Sea (Leck and Rodhe, 1991);

0.4 ± 0.3 nM from an Atlantic meridional transect (Kettle et al (2001); ~ 0.75 nM (up to 3nM) from the northeast



subarctic Pacific Ocean (Kiene et al., 2017), and a large gradient of ~ 0.8 to 3.3 nM in the temperate Atlantic (Gros et al., 2022).

Significant linear correlations between DMSa and MeSHa were found in both ASITs (ASIT-control $R^2 = 0.73$, $p_{value} < 0.001$, $y = 0.11x + 0.05$; ASIT-$O_3$ $R^2 = 0.81$, $p_{value} < 0.001$, $y = 0.43x - 0.13$. Figure S.3), indicative of their common metabolic production pathway from precursor DMSP (Figure 8). Indeed, significant correlations were also observed between MeSH and DMSP ($r^2$(ASIT-control) = 0.56, $p_{value} = 0.008$; $r^2$(ASIT-$O_3$) = 0.83, $p_{value} = 0.002$; Figure 8b). While the slopes of the linear regressions of MeSH/DMS were generally higher for the ASIT-$O_3$ particularly in EXP A (Fig 8a), the concentrations of total DMSP were roughly equivalent between ASIT-control and ASIT-$O_3$ across the 3 experiments (slope = 1.1, intercept = -0.6 nM, $R^2 = 0.91$) and the slope of MeSH to DMSP were similar in both ASITs. Therefore, ozone did not influence the conversion of DMSP to MeSH as it did for DMS.

### 3.4 Fluxes of DMS and MeSH in the ASITs

The net fluxes of DMS and MeSH were determined from the headspace data via Eq 3 (Sect 2.5) and time series of the fluxes measured in the ASIT-control and ASIT-$O_3$ over the 3 experiments are shown in Figure 9. Positive fluxes of DMS from the seawater to air were observed in all three experiments with the highest fluxes observed from the biologically productive frontal seawater incubation (EXP A: $F_{DMS} = 1.44 \pm 0.92$ ng m$^{-2}$ s$^{-1}$), moderate fluxes observed from the sub-Antarctic seawater (EXP B: $F_{DMS} = 0.51 \pm 0.39$ ng m$^{-2}$ s$^{-1}$) and lowest fluxes from the sub-Tropical water ($0.18 \pm 0.08$ ng m$^{-2}$ s$^{-1}$). MeSH fluxes were 6 to 8 times smaller but followed the same trends as DMS with fluxes of $0.17 \pm 0.06$ ng m$^{-2}$ s$^{-1}$ in the frontal waters, $0.08 \pm 0.03$ ng m$^{-2}$ s$^{-1}$ in the sub-Antarctic water sample and $0.04 \pm 0.03$ ng m$^{-2}$ s$^{-1}$ (Table 1).

In a companion study of DMSP/ DMS production and loss in incubations of sea surface microlayer samples on the same voyage biological consumption and dark production of DMS were the dominant processes, with irradiance having a net minor influence on DMSP/DMS cycling (Saint-Macary et al 2022). In the present study PTR-MS data were mostly collected during the nighttime (71%) which significantly limited the direct observation of the influence of PAR on DMS and MeSH production. However during EXP A when DMS and MeSH fluxes were highest, we observed that fluxes started to increase at night shortly before midnight and decreased from sunrise onwards. Furthermore, the pattern in DMS and MeSH fluxes did not follow the patterns in PAR measured in the ASITs experiments, with the highest PAR observed during the sub-Tropical water incubation in EXP C (max 1600 µmol m$^{-2}$ s$^{-1}$) which had the lowest DMS and MeSH fluxes. Likewise, no correlation ($r^2 < 0.03$, Table S.1) was found between the DMS and MeSH fluxes and seawater temperature in ASITs. It is unlikely that temperature variations within this range (13°C-18°C) had a significant effect on the Henry's Law partitioning ($C_{air}/C_{water}$) between the gas and aqueous phase for the VOCs studies (Sinha et al 2007). Overall, while irradiance and water temperature likely influence ocean-atmosphere fluxes on seasonal or latitudinal scales, changes in these environmental parameters did not appear to be major drivers of fluxes in these mesocosm scale experiments.

### 3.5 Relationships to biogeochemical properties of seawater samples

A range of seawater biogeochemical parameters were also analysed in the three ASIT experiments (Table 1, see Sellegri et al 2023 for methods). Many biogeochemical parameters, including Chl-a, nitrate, phosphate, CDOM,



DCHO and DAA exhibited similar or slightly greater in frontal waters relative to subtropical water, with both

exceeding concentrations in sub-Antarctic water (see Table 1). Conversely, silicate, iodide and iodate

concentrations showed the reverse, with lowest values in frontal waters and highest in sub-Tropical water sample.

These observations are consistent with the general classification of Sub-Antarctic seawaters as high nutrient low

chlorophyll (HNLC), and sub-Tropical waters being low in nutrients but replete in micro-nutrients, and the frontal

regions where these converge having high biological productivity. This high frontal biological productivity is

evident in the higher total phytoplankton biomass measured in the frontal seawater sample (51 mg C m$^{-3}$) in

comparison to the sub-Antarctic (12 mg C m$^{-3}$) and sub-Tropical samples (19 mg C m$^{-3}$).

In EXP A and C the phytoplankton biomass consisted of a greater proportion of nanophytoplankton (2 – 20 µm)

with the phytoplankton community (> 5 µm) comprised of dinoflagellates (~50 % of total C biomass) and diatoms

(~40 % of total C biomass), with a high proportion of large (> 20 µm) diatoms of the *Thalassiosira* genus. Slightly

higher phytoplankton volume and carbon were observed in the sub-Tropical seawater in EXP C, which was

dominated by larger (> 20 µm) diatoms including a high proportion of *Guinardia* and *Cylindrotheca* sp. In sub-

Antarctic water in EXP B, the phytoplankton community composition had relatively low total phytoplankton

volume and carbon, with smaller (10 - 20 µm) dinoflagellates dominating alongside a significant population of

diatoms of the *Chaetoceros* genus. Cell abundance of picophytoplankton (< 2 µm) and *Synechococcus* showed an

inverse relationship to the larger phytoplankton cell size groups, with a minimum in EXP A, and maxima in the

EXP B and C.; conversely bacterial abundance was greatest in EXP A (see Table 2).

For most paired samples the concentrations of biogeochemical parameters were similar in ASIT-control and

ASIT-O$_3$, and so the data from both ASITs was merged for each experiment, to determine relationships between

DMS and MeSH fluxes and bulk seawater biogeochemical properties (see Table 2). Only weak associations (R$^2$

< 0.2, Table S.2) were observed between Chl-a and concentrations of dissolved DMS and MeSH, and total DMSP,

whereas there were significant relationships between DMS, DMSP and MeSH with components of the

phytoplankton community, reflecting the variable content of DMSP of different phytoplankton groups (Keller et

al., 1989; Townsend and Keller, 1996). Figure 10 shows moderate to strong positive correlations between

nanophytoplankton number (cells mL$^{-1}$) and DMSP (R$^2$ = 0.52), DMSw (R$^2$ = 0.76), and MeSHw (R$^2$ = 0.60). The

nanophytoplankton size range can comprise various taxonomic groups including coccolithophores, diatoms,

dinoflagellates and cyanobacteria.

While moderate positive correlations were observed between DMSP, DMSw and MeSHw, with the biomass (mg

C m$^{-3}$) of diatoms (R² = 0.25 – 0.48), flagellates (R² = 0.34 – 0.62) and also bacterial abundance (cells/mL) (R² =

0.15 – 0.57), they correlated most strongly with the dinoflagellate biomass (R² = 0.6 – 0.70). Dinoflagellates are

high DMSP producers (Stefels et al 2007, Keller 1989), with some species having the capacity to directly cleave

DMSP to DMS (Nikki et al 2000, Wolfe & Steinke 1996). High DMS in frontal waters in this region was

previously associated with dinofagellates (Walker et al, 2016) and the contemporary study of Saint-Macary et al

(2023) also highlighted the importance of dinoflagellates in determining DMS and DMSP in the sea surface

microlayer. Conversely, large diatoms are considered relatively low DMSP producers and their dominance in the

sub-Tropical water in EXP C may explain why, despite higher overall phytoplankton Chl-a biomass, the fluxes of



DMS and MeSH were relatively low. Bacterial abundance was also moderately (5 – 20%) higher in the frontal
seawater which may reflect higher bacterial catabolism of DMSP to DMS and MeSH (Yoch, 2002) although
process rate measurements were not made.

The DMSw:DMSP ratio averaged ~ 0.06 across the three experiments with elevated ratios of 0.17 and 0.09
observed in the ASIT-control on day 2 of EXP A and day 2 of EXP B, respectively. Previous studies in this region
reported DMSw:DMSP ratios of 0.05 – 0.07, typical of phytoplankton blooms in which DMS production is
dominated by bacterially-mediated pathways, whereas ratios > 0.15 associated with large changes in DMSw are
indicative of additional phytoplankton-mediated DMSw production (Lizotte et al 2017).

In EXP A and B the net fluxes of DMS and MeSH exhibited a diel pattern with maxima during night-time (Fig
7), followed by a decline over the first few hours after sunrise, which may be attributed to enhanced photochemical
removal of DMS and MeSH via reaction with OH during daylight hours. However, the 40-minute residence time
in the ASITs headspace, and constant relationship of DMSa to DMSw, indicates minimal photochemical removal
(cf section 3.1). Instead, biological processes may have influenced the diurnal cycling of DMSP, DMS and MeSH
as these also exhibit diel patterns which may in turn contribute to diurnal changes in the net fluxes observed in
the ASITs experiments. For instance, the production and release of organic matter from phytoplankton cells
(Halsey & Jones, 2015 ), due to cell lysis from viral infection or grazing, may synchronise with the diel cycle, and
viral lysis predominantly occurs at night (Horas et al., 2018). Within phytoplankton cells, DMSP can represent a
significant fraction of the intracellular organic carbon (Thornton et al., 2014), and its release can be channelled
via exudation from phytoplankton cells, viral infection and cell lysis, or grazing ('sloppy feeding') (Stefels et al.,
470    2007).

The release of labile to semi-labile substrates such as DAA and DCHO follows similar patterns (Thornton, 2014;
Maron et al., 2022), and DMS fluxes were found to be strongly linked to the presence of DAA in both ASITs
(Table 2). Moreover, MeSH fluxes were strongly correlated to several phytoplankton groups, bacterial abundance,
and the concentration of DAA and DCHO in seawater (Table 2). This implies complex biological interactions
spanning from organic matter release to bacterial consumption and organic matter conversion. For heterotrophic
bacteria, DMSP is not only a source of energy but also provides sulfur. Bacteria produce MeSH and DMS while
degrading DMSP (Sun et al., 2016). The different intracellular pathways balance nutrient demands and MeSH and
DMS may be lost from bacterial cells by diffusion (Sun et al., 2016). In general, bacterial growth closely follows
phytoplankton bloom development and labile to semi-labile substrate availability (Amon et al., 2001; Shen and
Benner, 2020). The alignment of MeSH fluxes and bacterial abundance suggests coupling between bacterial
growth dynamics and DMSP degradation to MeSH, while the DMS fluxes depend on multiple source organisms
including bacteria and phytoplankton (Stefels et al., 2007; Sun et al., 2016).

### 3.6 Representativeness of the ASITs DMS and MeSH fluxes

DMS fluxes measured from the ASIT experiments, and also estimated from ambient concentrations (using the
two-layers method and nocturnal boundary layer method described sections 2.6 and 2.7) are compared to literature
values in Table 3. They computed DMS sea-to-air fluxes with a mean flux value of 12.9 $\pm$ 30.2 ng m$^{-2}$ s$^{-1}$. In Lana



et al., (2011), the authors determine DMS fluxes for the South-West Pacific region of around 5-7 ngS m$^{-2}$ s$^{-1}$, whereas Saint-Macary estimated mean fluxes of 4.44 ± 1.05 ng m$^{-2}$ s$^{-1}$ for sub-Tropical waters, 2.79 ± 0.83 ng m$^{-2}$ s$^{-1}$ for sub-Antarctic waters and 2.16 ± 0.46 ng m$^{-2}$ s$^{-1}$ for mixed waters. DMS fluxes determined during the Sea2Cloud and the SOAP voyages (Lawson et al 2020) for the same region using the nocturnal boundary layer accumulation method were 1-2 orders of magnitude greater than the fluxes in the ASIT experiments, but in line with fluxes estimated with the same method in the current study. Therefore the flux measurement method is important when comparing fluxes from one study to the other. This is expected as fluxes depend on wind speed, turbulence and dispersion which are limited in mesocosm and ASITs study. Overall, the broad range of regional DMS flux estimates may reflect different factors, including methodological assumptions. For example, flux measurements based on the nocturnal boundary layer accumulation method may be biased by long-range transport and vertical dilution which decouple ambient air from the underlying seawater. In addition seasonal differences in phytoplankton community composition, mixed layer depth, windspeed and temperature will also result in variability in fluxes.

A summary of the few studies that have reported air-sea fluxes of MeSH is provided in Table 4. Lawson et al. (2020) found fluxes of MeSH of 2.61 ng m$^{-2}$ s$^{-1}$ exceeding those measured during the current study by a factor of 30 but in line with the calculated flux from nocturnal boundary layer method. Kettle et al., (2001) and Leck and Rodhe, (1991) from the Baltic Sea also measured fluxes exceeding ours by a factor 4 to 20. However, MeSHw concentrations reported by Leck and Rodhe (1991) are one order of magnitude lower (0.16 nM) than the present study and the MeSH seawater concentrations reported by Kiene et al. (2017) (0.75 nM) are of the same order as those reported here. Therefore the overall lower MeSH fluxes observed in this study are not explained by lower seawater concentrations but, as for DMS, may be explained by differences in flux measurement methodologies.

In the ASIT-control experiments the average fluxes of MeSH compared to DMS (MeSH/(MeSH+DMS)) were 11%, 14% and 18% in the incubations of frontal, sub-Antarctic and sub-Tropical samples, respectively. These are consistent with the flux ratios determined via the nocturnal boundary layer accumulation method in the current study, with an average of 9%, and also during SOAP (14 - 24%, Lawson et al 2020) ; furthermore, they are within the range of ~ 5 – 20% determined in northern hemisphere (studies Leck & Rhode 1991, Kettle et al 2001, Kiene et al.,2017, Gros et al 2022) and also coastal bloom studies (Kilgour et al 2021, Novak et al 2021). Thus, while the absolute fluxes of DMS and MeSH reported via the different methods show a high variability there appears to be a fairly consistent relationship between the relative fluxes of MeSH and DMS across studies. Overall, these results suggest marine MeSH emissions can significantly contribute to the total atmospheric sulfur budget.

**3.7 Biological-flux relationships for modelling**

Sea surface Chl-a concentrations are often used as a generic proxy for phytoplankton biomass in the marine environment and are used in conjunction with other physical and biogeochemical variables in model parameterizations of DMS fluxes (Lana et al 2011, Simo & Dachs et al 2002, Aumont et al 2002). However, DMSw and MeSHw were not significantly linearly correlated with Chl-a in the ASITs, consistent with previous studies (Bell et al., 2021; Galí et al., 2018; Leck et al., 1990; Townsend and Keller, 1996; Kettle et al., 1999; Simó et al., 1995), pointing to the need for a more



appropriate biological tracer. Instead, a relationship of DMS and MeSH fluxes upon nanophytoplankton cell abundance was derived based on the merged data from all 3 ASIT experiments and two ASITs (ASIT-control and ASIT-$O_3$):

$$[F_{DMS\text{-}ASITs}] = 1.35 \times 10^{-3} \text{ [nanophyto]} - 1.06 \ (r = 0.58, \ p\_val < 0.001) \tag{Eq. 9}$$

$$[F_{MeSH\text{-}ASITS}] = 1.61 \times 10^{-4} \text{ [nanophyto]} - 0.09 \ (r = 0.65, \ p\_val < 0.001) \tag{Eq. 10}$$

These general relationships given by Equation 11 and Equation 12 are presented to facilitate the prediction of DMS and MeSH fluxes in chemistry transport models as a function of phytoplanktonic groups available as satellite product (see Uitz et al. 2006 for nanophytoplankton), although they do not consider other factors that modulate the fluxes of DMS and MeSH to the atmosphere (see section 3.5.3).

In order to take wind speed and turbulence into account, fluxes may be predicted from DMSw and MeSHw. Therefore, relationships were also derived for DMSw and MeSHw with nanophytoplankton abundance as follows:

$$[DMS_{W\text{-}ASITs}] \text{ (nM)} = 4.02 \ 10^{-3} \text{ [nanophyto] (cell mL}^{-1}) - 2.41 \ (r = 0.87, \ p\_val < 0.001) \tag{Eq. 11}$$

$$[MeSH_{W\text{-}ASITs}] \text{ (nM)} = 9.38 \ 10^{-4} \text{ [nanophyto](cell mL}^{-1}) - 0.80 \ (r = 0.78, \ p\_val < 0.001) \tag{Eq. 12}$$

In ambient seawater (sampled via workboat and CTD, Figure S.8), DMSw was also significantly correlated to nanophytoplankton cell abundances with the following relationship:

$$[DMS_{w\text{-}ambiant}] \text{ (nM)} = 1.26 \ 10^{-3} \text{ [nanophyto] (cell mL}^{-1}) + 0.62 \ (r = 0.63, \ p\_val < 0.001) \tag{Eq. 13}$$

Hence, we confirm that the relationships observed in ambient seawater sampled during the voyage have coefficients that are close to those found in the ASITs seawater.

### 4. Conclusion

Fluxes of MeSH and DMS were determined over the Southern Ocean during the Sea2Cloud campaign on the R/V Tangaroa using Air Sea Interface Tanks (ASITs). Seawater of differing origins were studied: frontal, sub-Antarctic and sub-Tropical waters, with paired experiments conducted with differing headspace ozone concentrations. Good agreement was observed between DMSa and DMSw in line with that expected from thermodynamical equilibrium being achieved in the ASIT's headspaces. We found that fluxes measured in the ASITs could be accurately predicted from their seawater concentrations, SST and an equivalent wind speed, using empirical relationships used in the modelling community (the two-layer model). Overall, the ASITs studies enabled the relationships between emissions and seawater biogeochemistry and plankton community dynamics under near-natural light and temperature conditions to be explored in situ. Fluxes of DMS and MeSH showed significant relationship with nanophytoplankton cell abundance, and this was used to generate parameterizations for both DMS and MeSH flux, and also DMSw using a large dataset. Recent decades have seen an expansion in the use of more sophisticated bio-optical remote sensing products



that can extract information on phytoplankton functional types, size classes and taxonomic composition (e.g., Gantt et al 2009, Alvain et al 2005); these can be combined with the derived relationships to provide regional estimates of DMS and MeSH fluxes using thermodynamic and

kinetic empirical relationships as in the current study using models such as COARE for DMS and theoretical gas-liquid phase equations for MeSH.

The relative production of MeSH to DMS in biologically productive frontal waters appeared to be affected by enhanced ozone. Due to limited data, the nature of the ozone interaction remains unclear

however one possible explanation is that ozone limited processes involved in the release of intracellular DMSP perhaps by suppressing bacterial/viral lysis of phytoplankton cells, thus limiting conversion to DMS. The ozone concentrations tested were within the typical range in the ambient atmosphere in this region, pointing to a potentially important metabolic influence of ozone that needs to be further investigated.


**Data availability**
Datasets reported in this manuscript are available at the Sea2Cloud project data repository at
https://sea2cloud.data-terra.org/en/catalogue/

**Author contributions statement**

All authors contributed to the design of the experiment, collection of data and revision of the manuscript (E.D., C.S. L., A.C.,  K.Sellegri. N.B., M.P., A. S-M., J. H.). Data analysis and preparation of the manuscript were performed by M.R., E.D., C.S. L., T. B., A.C., M. H., A. S.L., and K.Sellegri. N.B. conceived the ASITs. C.S.L., S.D., K.Safi and A. E., T. B., A. S-M., A.M. provide phytoplankton data.

**Competing interests**

The authors declare no competing interests.

**Author contributions statement**
This paper is corresponding to the special issue Sea2Cloud Sea2Cloud (ACP/OS inter-journal SI).

**Acknowledgements:** This research received funding from the European Research Council (ERC)
under the Horizon 2020 research and innovation programme (Grant agreement number - 771369), and was supported by New Zealand Scientific Strategic Funding via the NIWA Climate, Atmosphere and Hazards Centre. Sea2Cloud is endorsed by SOLAS.





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





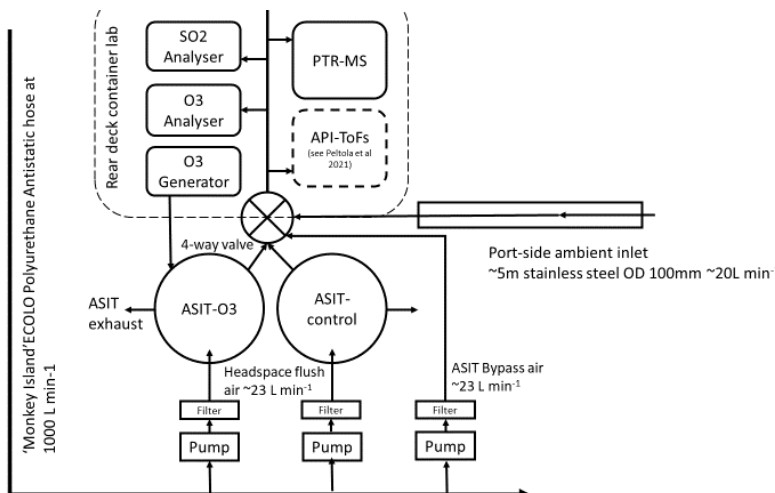

Figure 1: Schematic of the ASITs experimental set-up.

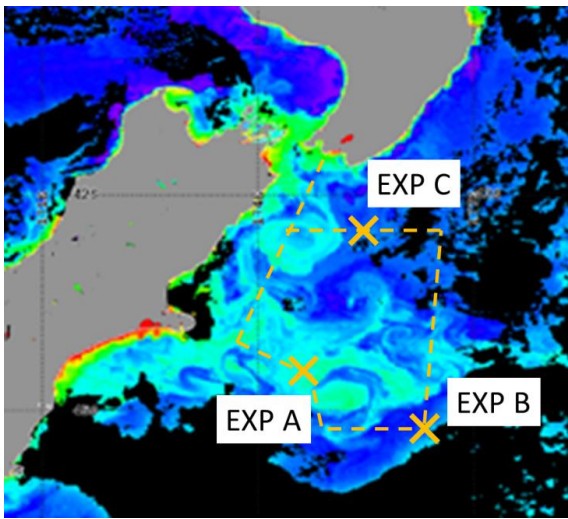

Figure 2: Location of the 3 ASIT experiments (EXP A- frontal, EXP B - subantarctic, EXP C - subtropical). Overlain on a satellite image of ocean colour (*b_bp443*) on 14/3/20, showing the variability and structure of blooms
975 along the Chatham Rise (Image data generated by the Visible Infrared Imaging Radiometer Suite (VIIRS) onboard the Suomi National Polar-orbiting Partnership (SNPP) satellite; data courtesy of NOAA / NESDIS Center for Satellite Applications and Research).



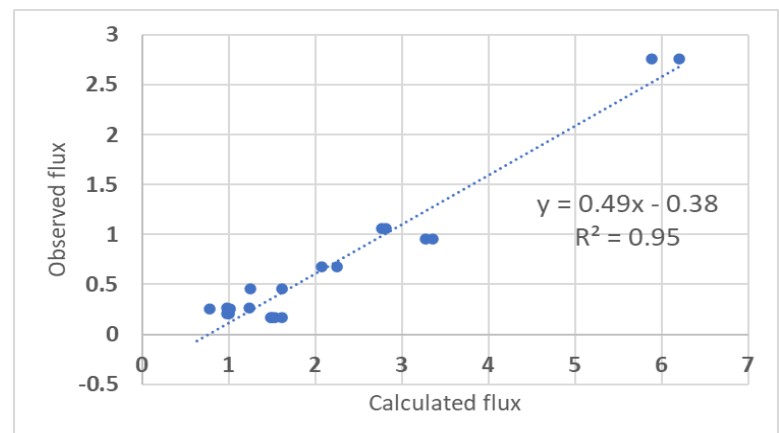

**Figure 3: Correlations between DMS fluxes calculated from equation 5 ($F_c$) and DMS fluxes measured in both ASITs using equation 3 (Section 2.5).**

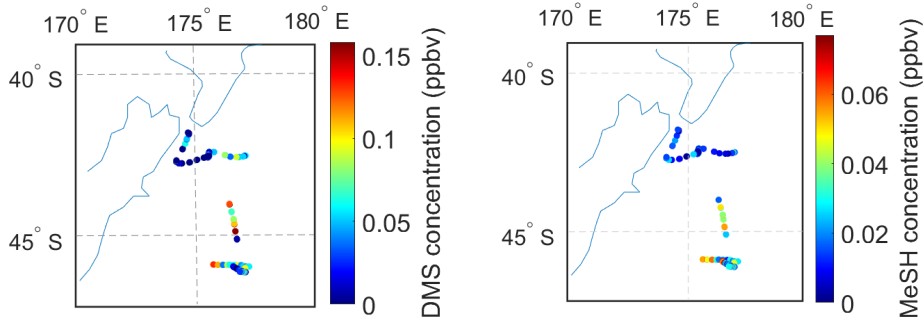

**Figure 4: Ambient MBL concentrations of DMS and MeSH in ppbv.**

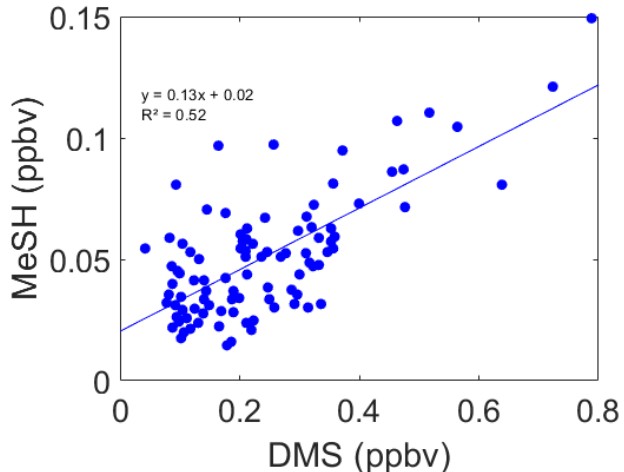

**Figure 5: Correlation between ambient DMS and MeSH concentration (ppb) in the MBL.**



985

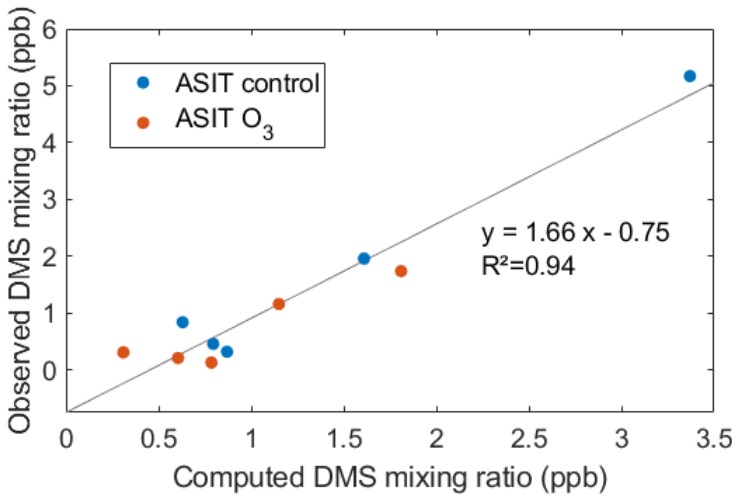

**Figure 6: DMS headspace mixing ratios (ppbv) measured in ASITs headspace versus DMS headspace mixing ratios (ppbv) computed from DMS concentrations in the seawater (in nM) using Henry's law.**

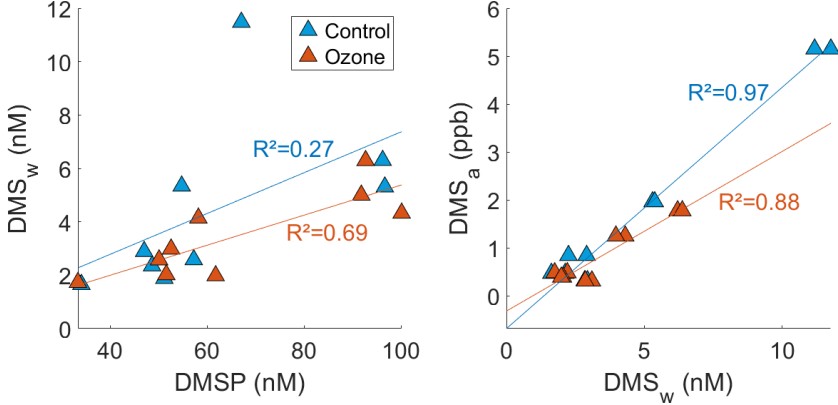

990 **Figure 7. (a) Dissolved DMS concentration in the seawater (nM) as a function of DMSP concentrations in the seawater (nM); (b) DMS headspace concentrations as a function of dissolved concentration in the ASIT seawater.**



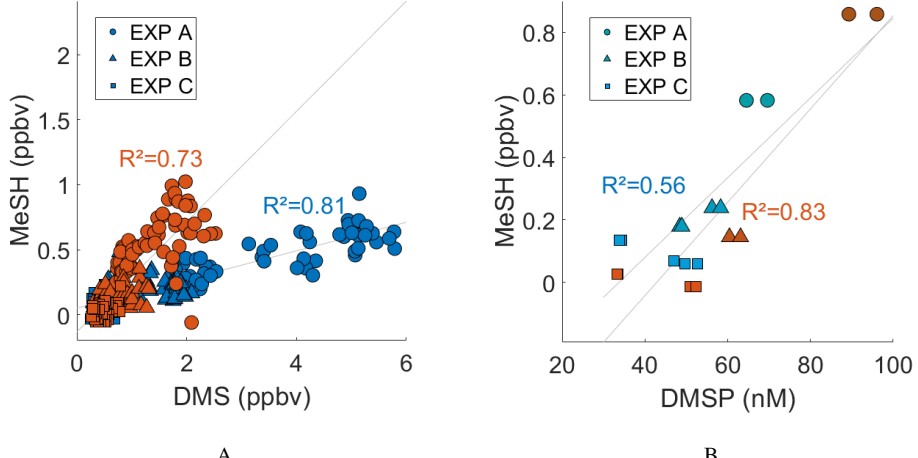

A                                    B

**Figure 8: A) Concentration of DMSa vs MeSHa in ppbv. EXP A is represented by circles, B by triangles and C by squares. ASIT-control = blue color, ASIT-O$_3$ = orange color. r²(ASIT-control) = 0.73, p$_{value}$ < 0.001, y = 0.11x + 0.05; r²(ASIT-O$_3$) = 0.81, p$_{value}$ < 0.001, y=0.43x -0.13. B) Concentrations of MeSH (ppbv) vs DMSP concentration (nM) in both ASITs**

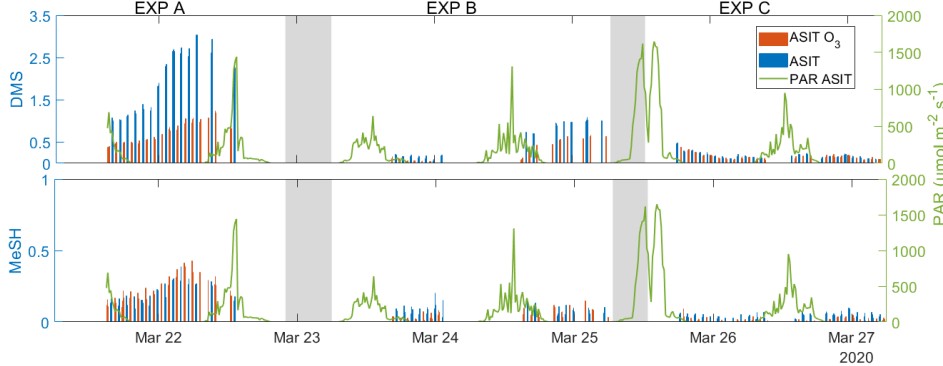

995

**Figure 9: DMS and MeSH fluxes in ng m$^{-1}$ s$^{-1}$ from ASIT control (blue dots) and ozone ASIT (orange dots). Photosynthetically Active Radiation (PAR, green line) in µmol m$^{-1}$ s$^{-1}$.**



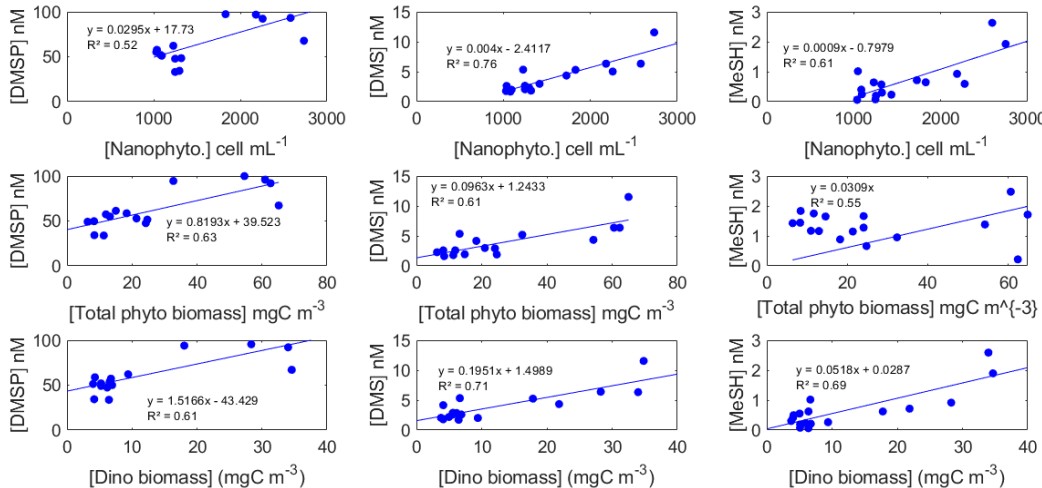

**Figure 10: Correlation plots of DMSP, DMS and MeSH concentrations in the ASITS (ASIT-control and ASIT-O₃ combined) against nanophytoplankton (upper row), total phytoplankton carbon biomass (middle row) and dinoflagellate carbon biomass (lower row).**

| | EXP A Frontal | EXP B Sub-Antarctic | EXP C Sub-Tropical |
|---|---|---|---|
| **Chlorophyll biomass** | | | |
| Total Chl-a (mg m$^{-3}$) | 1.87 ± 0.34 | 0.54 ± 0.12 | 1.88 ± 0.30 |
| Chl-a >20 um (mg m$^{-3}$) | 0.98 ± 0.23 | 0.13 ± 0.06 | 1.17 ± 0.25 |
| Chl-a 2-20 um (mg m$^{-3}$) | 0.29 ± 0.03 | 0.14 ± 0.03 | 0.25 ± 0.06 |
| Chl-a <2 um (mg m$^{-3}$) | 0.26 ± 0.08 | 0.18 ± 0.03 | 0.29 ± 0.05 |
| **Phytoplankton Size Class** | | | |
| Nanophytoplankton (cells mL$^{-1}$) | 2222 ± 399 | 1193 ± 116 | 1227 ± 157 |
| Picophytoplankton (cells mL$^{-1}$) | 8681 ± 2161 | 15348 ± 1123 | 16060 ± 7334 |
| Synechococcus (cells mL$^{-1}$) | 32514 ± 5695 | 63417 ± 5870 | 41314 ± 7574 |
| Bacteria (cells mL$^{-1}$) | 3354983 ± 1076114 | 2276892.56 ± 155596 | 2435937.72 ± 137492 |
| **Phytoplankton biomass** | | | |
| Total > 5um (mg C$^{-3}$) | 51.0 ± 15.4 | 12.1 ± 4.3 | 19.1 ± 7.3 |
| Dinoflagellates (mg C$^{-3}$) | 17.7 ± 14.9 | 6.5 ± 1.7 | 5.1 ± 1.1 |
| Diatoms (mg C$^{-3}$) | 12.8 ± 11.9 | 4.2 ± 4.4 | 12.4 ± 7.2 |
| Flagellates (mg C$^{-3}$) | 2.0 ± 0.9 | 1.5 ± 0.4 | 1.6 ± 0.2 |



| Dissolved and Particulate Organics | | | |
|---|---|---|---|
| Particulate Nitrogen (mg $C^{-3}$) | 56 ± 6 | 24 ± 4 | 31 ± 4 |
| Particulate Carbon (mg $C^{-3}$) | 394 ± 46 | 119 ± 17 | 204 ± 51 |
| CDOM (ppbv) | 0.23 ± 0.02 | 0.16 ± 0.01 | 0.24 ± 0.01 |
| DCHO | 1013 ± 452 | 331 ± 287 | 684 ± 123 |
| Dissolved amino-acids (DAA) (nmol/L) | 459 ± 153 | 397 ± 78 | 565 ± 365 |
| Total amino-acids (DAA) (nmol $L^{-1}$) | 2295 ± 927 | 1149 ± 471 | 1273 ± 678 |
| Iodide (nmol $L^{-1}$) | 9.70 ± 2.77 | 21.93 ± 5.11 | 30.85 ± 5.44 |
| Iodate (nmol $L^{-1}$) | 141.58 ± 23.22 | 204.03 ± 44.32 | 388.15 ± 53.67 |
| DOC (μM) | 96.8 | 80.4 | 93.3 |
| DMS (nmol $L^{-1}$) | 6.45 ± 2.58 | 3.16 ± 1.30 | 2.19 ± 0.59 |
| DMSP (nmol $L^{-1}$) | 90.73 ± 11.98 | 55.12 ± 5.00 | 44.97 ± 8.91 |

**Table 1: Mean concentrations (± 1 S.D.) for biogeochemical parameters during the three ASIT experiments.**

| r (p_value) | DMSa (n=18) | MeSHa (n=18) |
|---|---|---|
| Picophytoplankton (cells $mL^{-1}$) (n=16) | -0.36 (<0.001) | -0.24 (<0.001) |
| Prokaryotic pico - Syne, PrKS (cells $mL^{-1}$) (n=16) | -0.42 (<0.001) | **-0.65** (<0.001) |
| Nanophytoplankton (cells $mL^{-1}$) (n=16) | **0.73** (<0.001) | **0.86** (<0.001) |
| Dinoflagellates (mgC $m^{-3}$) (n=18) | **0.74** (<0.001) | **0.83** (<0.001) |
| Diatoms (mgC $m^{-3}$) (n=18) | **0.45** (<0.001) | **0.56** (<0.001) |
| Flagellates (mgC $m^{-3}$) (n=18) | **0.72** (<0.001) | **0.63** (<0.001) |
| Bacteria (cells $mL^{-1}$) (n=18) | 0.15 (<0.001) | **0.80** (<0.001) |
| Chl-a (mg $m^{-3}$) (n=18) | -0.02 (0.148) | 0.25 (<0.001) |
| $DMS_w$ (nM) (n = 18) | **0.93** (<0.001) | **0.63** (<0.001) |
| DMSP (nM) (n = 18) | 0.19 (<0.001) | **0.52** (<0.001) |





| | | |
|---|---|---|
| DOC (µM) SUB (n=8) | 0.17 (<0.001) | 0.37 (<0.001) |
| PN (µg N L$^{-1}$) (n=18) | 0.49 (<0.001) | **0.75** (<0.001) |
| PC (µg C L$^{-1}$) (n=18) | 0.43 (<0.001) | **0.71** (<0.001) |
| DCHO (nmol L$^{-1}$) SUB | 0.53 (<0.001) | **0.72** (<0.001) |
| DAA (sum in nmol L$^{-1}$) SUB | 0.67 (<0.001) | **0.75** (<0.001) |
| iodide (nM) | **-0.52** (<0.001) | **-0.69** (<0.001) |
| iodate (nM) | -0.40 (<0.001) | **-0.53** (<0.001) |

1005    **Table 2: Correlation between fluxes and seawater biogeochemical variables derived from all samples from the ASIT experiments. Bold values indicate strong correlations (r > 0.5 or r < -0.5), with negative correlation shown in blue, and positive correlations in red.**

| DMS Flux (ng m$^{-2}$ s$^{-1}$) | Seawater origin | Method | Reference |
|---|---|---|---|
| 1.44 ± 0.92<br>0.51 ± 0.39<br>0.18 ± 0.08 | Frontal waters<br>Sub-Antarctic water<br>Sub-Tropical water | ASIT-Control | This study |
| 0.67 ± 0.26<br>0.22 ± 0.24<br>0.15 ± 0.07 | Frontal waters<br>Sub-Antarctic water<br>Sub-Tropical water | ASIT-O$_3$ | |
| 20.5 ± 15.9 | South-West Pacific -Chatham Rise (~44°S) | Nocturnal Boundary Layer Accumulation method | This study |
| 9.1 ± 5.3 | South-West Pacific - Chatham Rise (~42 - 46°S) | | Lawson et al (2020) |
| 1.64 – 7.89 | South-West Pacific - Chatham Rise (~42 - 46°S) | COARE | Saint-Macary et al., (2023) |
| ~5.74 – 7.89 | Sub-Antarctic & Antarctic zone (58 - 42°S) | 2-layer model | Zhang et al (2021) |
| ~3.58 – 5.74 | South sub-Tropical zone (42 - 15°S) | | |
| 0.71 – 61.72 | South-West Pacific – Ross Sea (49 - 76.5° S) | 2-layer model | Kiene et al (2017) |
| 2.15 – 43.1 | Southern Ocean | Eddy Covariance | Bell et al., 2015 |

**Table 3: Average (± std. deviation) of DMS fluxes in ng m$^{-2}$ s$^{-1}$ for the three experiments in the ASITs and comparison with literature**

1010    **values for from the Southern hemisphere.**





| Average F(MeSH) (ng m$^{-2}$ s$^{-1}$) | Average F(MeSH) % of total | Method/location | Reference |
|---|---|---|---|
| 0.17 ± 0.06 0.08 ± 0.03 0.04 ± 0.03 | 11% 14% 18% | ASIT-Control Frontal (EXP A) Sub-Antarctic (EXP B) Sub-Tropical (EXP C) | This study |
| 0.21 ± 0.09 0.04 ± 0.04 0.01 ± 0.02 | 24% 15% 6% | ASIT-O3 Frontal (EXP A) Sub-Antarctic (EXP B) Sub-Tropical (EXP C) | |
| 1.9 ± 1.4 | 9% | Nocturnal Accumulation method | This study |
| 2.61 (2.5 – 4.2) | 14 – 24% | | Lawson et al. (2020) |
| 0.21 (0.10-0.31) ppt m$^{-2}$ s$^{-1}$ | | Eddy covariance, North Pacific coastal waters | Novak et al. (2021) |

**Table 4: Average (± std. deviation) of MeSH fluxes in ng m$^{-2}$ s$^{-1}$ for the three experiments in the ASITs and comparison with literature values.**