# Peer review of "Air-Sea fluxes of dimethyl sulphide and methanethiol in the South-West Pacific"

_EGUsphere, 2023_

## Referee Comment (RC1)

**Review of MS Egusphere 2023-516.**

Neither in its form nor in its scientific content is this manuscript suitable for publication in Atmospheric Chemistry and Physics (ACP).

**In its form.** My overall feeling is that this manuscript has been written in a rush and, contrary to the author contributions statement, all authors very likely did not contribute to its writing, review and editing, e.g.:

- 2 different titles: in the main text the MS is entitled "Air-Sea fluxes of dimethyl sulphide and methanethiol in the South-West Pacific" whereas in the supplementary it is entitled "Sea-Air fluxes of dimethyl sulphide and methanethiol from mesocosm studies of natural seawaters from the South-West Pacific".
- Reference list: About 17 references cited in the reference list are not cited in the main text. About 12 references cited in the main text are not listed in the reference list, including the recent contributions of the corresponding author (Sellegri et al., 2022; Sellegri et al., 2023)!
- Data displayed in Fig. 4 and Fig. 5 are inconsistent. DMS mixing ratios are 10 times lower in Fig. 4 than in Fig. 5.
- Fig. S.1: Although RMA regression instead of least-squares regression has been used in Fig. S.1 (shown below), I doubt that the authors can conclude that "the correlations between DMS in air and water did not differ between the ASIT-control and ASIT-O3". The positive blue regression (applied to ASIT-control isn't it?) is in fact driven by only two data points. In terms of slope, intercept and r2 values, why are different numbers provided in text and Fig. S.1? Why are the r and r2 values the same? Was RMA applied throughout the text?

[Figure]

ASIT-control : slope = 0.28 ± 0.41; intercept = -0.31 ± 0.08; r = 0.82

ASIT-O₃: slope = 0.21 ± 0.32; intercept = -0.17 ± 0.08; r = 0.61

**Figure S. 1: Reduced major axis (RMA) of measured ASITs DMS (seawater) vs. DMS (headspace).**

**In its scientific content.**

This work makes insufficient use of DMS state-of-the-art. References to recent studies are missing in the introduction. The Lana et al. (2011) climatology of marine DMS has been updated recently (https://doi.org/10.5194/essd-14-2963-2022). It now incorporates more than 1000 data points collected by Bell et al. (2015) in the same area than that investigated during the Sea2Cloud campaign (see below) where only a few tens of samples were analyzed for DMSw in the framework of the ASIT experiments. You should make this clear in the introduction.

**DMS Database Output**

[Figure]

| Limits | Minimum | Maximum |
|---|---|---|
| Date | 1972-03-11 | 2019-04-15 |
| Latitude | -47 | -40 |
| Longitude | 173 | 178 |
| Month(s) | all | |

[Figure]

| Data Record Statistics | |
|---|---|
| Number of records | 1012 |
| Minimum DMS value | 1.10 |
| 95th Percentile DMS value | 17.05 |
| Maximum DMS value | 33.63 |
| Mean DMS Value | 7.28 |
| Median DMS Value | 5.83 |
| DMS Standard Deviation | 4.77 |

Download This Data Set
Please read the data file description

[Figure]

| Cont. Num | Contributor | Platform | Region | Num Samples | Start | End | Reference(s) |
|---|---|---|---|---|---|---|---|
| 247 | Bell | R/V Tangaroa | Southern Ocean | 1012 | 2012-02-15 | 2012-03-06 | Bell et al., 2015 |

In the introduction, it is also stated that "In the surface ocean, DMS is produced from the degradation of dimethylsulfoniopropionate (DMSP), which is produced by marine macroalgae, phytoplankton or bacteria (Bentley and Chasteen, 2004; Kloster et al., 2006; Novak and Bertram, 2020). To form DMS, the DMSP undergoes reactions catalysed by DMSP lyase, (Taylor and Visscher, 1996; Steinke et al., 1996; Kiene, 1996a) and non-enzymatic pathways of demethylation (Bentley and

Chasteen, 2004). Furthermore, DMS can be produced by the biological reduction and oxidation of DMSP and also abiotically by light-dependent reactions (McNabb and Tortell, 2021 and references therein)". According to the very recent review of Shaw et al. (2022, Shaw, D.K., Sekar, J. and Ramalingam, P.V. Recent insights into oceanic dimethylsulfoniopropionate biosynthesis and catabolism. Environmental Microbiology (2022) 24(6), 2669–2700. doi:10.1111/1462-2920.16045), the existence of a "non-enzymatic pathways of demethylation (Bentley and Chasteen, 2004)" is highly unlikely as is "DMS production by the biological reduction and oxidation of DMSP". I acknowledge that the metabolism and catabolism of DMSP is complex as shown below but, unfortunately, the present manuscript does not make any advance in this field. The importance of nanophytoplankton in DMSP synthesis has been established 20 years ago.

[Figure]

**Fig. 4.** Microbial biosynthesis and cycling of DMSP and DMS. AcuH, acryloyl-CoA hydratase; AlmA1, DMSP lyase; CCN, cloud condensation nuclei; Ddds, various DMSP lyases; DdhA, dimethylsulfide dehydrogenase; DmdA, DMSP demethylase; DmdB, MMPA-CoA ligase; DmdC, MMPA-CoA dehydrogenase; DmdD, methylthioacryloyl-CoA hydratase; DmoA, dimethylsulfide monooxygenase; DMS, dimethylsulfide; DMSO, dimethyl sulfoxide; DMSOR, dimethyl sulfoxide reductase; DMSP, dimethylsulfoniopropionate; DMSOP, dimethylsulfoxonium propionate; DsyB, DSYB; MddA, MeSH S-methyltransferase; MegL, Methionine γ-lyase; MeSH, methanethiol; MMPA, methylmercaptopropionate; MmtN, Met-methylating enzymes; MTO, MeSH oxidase; Tmm, trimethylamine monooxygenase; TpMMT, methylthiohydroxybutryrate SAM-dependent methyltransferase.

[Figure]

**Fig. 3.** Biochemical pathways for dimethylsulphoniopropionate catabolism. In the cleavage pathways, several DMSP lyases DddL, DddP, DddQ, DddW, DddK, DddX, DddY or algal AlmA1 catabolizes DMSP to acrylate with the release of dimethyl sulfide (DMS), and acrylate is then converted to 3-hydroxypropionate (3HP) by the action of AcuN and AcuK, whereas the DMSP lyase DddD converts DMSP to 3HP. 3HP is then converted to malonate semi-aldehyde (Mal-SA) and then acetyl-CoA by DddA and DddC, respectively. An acrylate-CoA ligase (PrpE), an acryloyl-CoA reductase (AcuI) and AcuH are also involved in the cleavage pathway. The DMSP demethylation pathway is catalysed by the DMSP demethylase (DmdA), MMPA-CoA ligase (DmdB), MMPA-CoA dehydrogenase (DmdC), and either the MTA-CoA hydratase (DmdD) or acrylate utilization hydratase (AcuH). In the oxidation pathway, DMSP is oxidized to dimethylsulfoxonium propionate (DMSOP). However, enzyme involved in this pathway is unknown.

I agree with Dr. Dong's comment. The sea-air fluxes of DMS and CH3SH measured in the ASIT tanks by no means can be compared with literature values reported in Table 3, because the turbulence very likely is considerably lower in the tanks than in the reality. By no means can this manuscript be entitled "Air-Sea fluxes of dimethyl sulphide and methanethiol in the South-West Pacific", and the following sentences "Air-sea fluxes of dimethyl sulphide (DMS) and methanethiol (MeSH) from surface seawater in the remote Southern Pacific Ocean were measured in three Air-Sea Interface Tank (ASIT) experiments during the Sea2Cloud voyage in March 2020. The measured fluxes of 0.78 ± 0.44 ng m-2 s-1 and 0.05 ± 0.03 ng m-2 s-1 for DMS and MeSH, respectively, varied between experiments reflecting the different water mass types investigated, with lowest fluxes with subtropical water and highest with biologically-active water with sub-Tropical water and highest from the sub-Tropical Front" be put forward in the abstract.

Another major flaw is the role attributed to ozone, i.e. "The experiments also determined the influence of elevated ozone, with one ASIT headspace amended with 10 ppbv ozone while the other provided an unamended control. Elevated ozone resulted in a decrease in DMS flux, corresponding to decreased conversion of dimethylsulfoniopropionate (DMSP) to DMS in the seawater". Concentrations of DMSP and DMS in seawater are displayed in Fig. S.5 and Fig. S.2, respectively (reproduced below). Although the poor quality of both figures makes the comparison difficult to be established, it seems that it is in ASIT-control not in ASIT-O3 that a major change (an increase in this case) in the DMS-to-DMSP ratio took place in EXP-A by Mar 22. How can one trust that ozone is an inhibitor of the conversion of DMSP to DMS in seawater when the demonstration relies on a single observation? This process should have been investigated in the laboratory under more controlled conditions and in replicates.

[Figure]

Figure S.5: Concentrations of DMSP in seawater (nM).

[Figure]

Figure S.2: Concentration of DMS in seawater (nM) and air (ppbv) in ASIT-control (blue) and ASIT-O₃ (orange). ~20min average DMS headspace mixing ratios (ppbv, dots ) in ASIT-control (blue) and ASIT-O3 (orange) and dissolved DMS in ASITs seawater samples(nM, triangles).

This manuscript should be rejected in its present form.

---

## Author Comment (AC1)

**Answers to comments on "Relating dimethyl sulphide and methanethiol fluxes to surface biota in the South-West Pacific using shipboard mesocosms" article submitted in *ACP*.**

Manon Rocco[1,2*], Erin Dunne[3], Alexia Saint-Macary[4,7], Maija Peltola[1], Theresa Barthelmeß[5], Neill Barr[4], Karl Safi[4A], Andrew Marriner[4], Stacy Deppeler[4], James Harnwell[3], Anja Engel[5], Aurélie Colomb[1], Alfonso Saiz-Lopez[6], Mike Harvey[4,†], Cliff S. Law[4,7], and Karine Sellegri[1]

[1] Université Clermont Auvergne, CNRS, Laboratoire de Météorologie Physique (LaMP), 63000, Clermont-Ferrand, France
[2] Now at Instituto de Astronomia, Geofísica e Ciências Atmosféricas/IAG - Universidade de São Paulo - Laboratório de Estudos Ambientais/IPA, São Paulo, Brazil
[3] CSIRO Environment, Aspendale, Australia
[4] National Institute of Water and Atmospheric Research, Wellington, New Zealand
[4A] National Institute of Water and Atmospheric Research, Hamilton, New Zealand
[5] GEOMAR Helmholtz Centre for Ocean Research Kiel, Kiel, Germany
[6] National Center for Atmospheric Research, NCAR, Boulder, USA
[7] Department of Marine Science, University of Otago, New Zealand
[†] Deceased

*Correspondence to*: Karine Sellegri (karine.sellegri@uca.fr); Manon Rocco (rocco.manon@gmail.com)

**#Reviewer 1:**

Neither in its form nor in its scientific content is this manuscript suitable for publication in Atmospheric Chemistry and Physics (ACP). In its form. My overall feeling is that this manuscript has been written in a rush and, contrary to the author contributions statement, all authors very likely did not contribute to its writing, review and editing, e.g.:

- 2 different titles: in the main text the MS is entitled "Air-Sea fluxes of dimethyl sulphide and methanethiol in the South-West Pacific" whereas in the supplementary it is entitled "Sea-Air fluxes of dimethyl sulphide and methanethiol from mesocosm studies of natural seawaters from the South-West Pacific".

We modified the title of the main text to : "Relating dimethyl sulphide and methanethiol fluxes to surface biota in the South-West Pacific using shipboard mesocosms".

- Reference list: About 17 references cited in the reference list are not cited in the main text. About 12 references cited in the main text are not listed in the reference list, including the recent contributions of the corresponding author (Sellegri et al., 2022; Sellegri et al., 2023)!

The reference list has been revised and corrected.

- Data displayed in Fig. 4 and Fig. 5 are inconsistent. DMS mixing ratios are 10 times lower in Fig. 4 than in Fig. 5.

Figure 4 has been revised and updated. Figure 5 is right.

- Fig. S.1: Although RMA regression instead of least-squares regression has been used in Fig. S.1 (shown below), I doubt that the authors can conclude that "the correlations between DMS in air

and water did not differ between the ASIT-control and ASIT-O3". The positive blue regression (applied to ASIT-control isn't it?) is in fact driven by only two data points. In terms of slope, intercept and r2 values, why are different numbers provided in text and Fig. S.1? Why are the r and r2 values the same? Was RMA applied throughout the text?

We apologise for the supplementary figures and text not being sufficiently checked. The Figure S.1 has been checked and revised. RMA was not applied throughout the text but applied on focused analysis to strengthen our hypothesis.

Given the uncertainty on each slope and intercept, the RMA analysis tells us that the slopes can not be distinguished from each other (NEW VALUES:
ASIT-control : slope = 0.51, interval of confidence = [0.45 0.58]; intercept = -0.72, interval of confidence= [-1.05 -0.42]; r = 0.98
ASIT-$O_3$: slope = 0.36, interval of confidence = [0.27 0.49]; intercept = -0.42, interval of confidence= [-0.88 -0.09]; r = 0.94) and therefore the relationship between DMS in air and DMS in the seawater is not different between the ASIT-control and ASIT-ozone.

The RMA regression without 2 outliers of the ASIT-control shows very similar regression slopes than the one with outliers (Figure S1.B, in the ASIT-control : slope = 0.51, interval of confidence = [0.31 0.87]; intercept = -0.69, interval of confidence= [-1.88 -0.09]; r = 0.93.)

[Figure]

Therefore the conclusion of indistinguishable DMSa/DMSw between ASIT-control and ASIT-ozone is also valid when excluding outliers and not only driven by a few high value points.

- This work makes insufficient use of DMS state-of-the-art. References to recent studies are missing in the introduction. The Lana et al. (2011) climatology of marine DMS has been updated recently (https://doi.org/10.5194/essd-14-2963-2022). It now incorporates more than 1000 data points collected by Bell et al. (2015) in the same area as that investigated during the Sea2Cloud campaign (see below) where only a few tens of samples were analyzed for DMSw in the framework of the ASIT experiments. You should make this clear in the introduction.

The updated version of the Lana climatology is now mentioned in the introduction. This version is very recent (2022) and therefore was not available at the start of writing the present paper, which explains why it was missing. We now include it. However, we contest the reviewer's assessment that our

introduction makes insufficient use of DMS state of the art . We use 32 different references, among which many are from the last 5 years (Novak and Bertram, 2020; McNabb and Tortell, 2021; Lawson et al., 2020; Kilgour et al., 2021; Wang et al., 2020;Galí et al., 2018; Bell et al., 2021).

- In the introduction, it is also stated that "In the surface ocean, DMS is produced from the degradation of dimethylsulfoniopropionate (DMSP), which is produced by marine macroalgae, phytoplankton or bacteria (Bentley and Chasteen, 2004; Kloster et al., 2006; Novak and Bertram, 2020). To form DMS, the DMSP undergoes reactions catalysed by DMSP lyase, (Taylor and Visscher, 1996; Steinke et al., 1996; Kiene, 1996a) and non-enzymatic pathways of demethylation (Bentley and Chasteen, 2004). Furthermore, DMS can be produced by the biological reduction and oxidation of DMSP and also abiotically by light-dependent reactions (McNabb and Tortell, 2021 and references therein)". According to the very recent review of Shaw et al. (2022, Shaw, D.K., Sekar, J. and Ramalingam, P.V. Recent insights into oceanic dimethylsulfoniopropionate biosynthesis and catabolism. Environmental Microbiology (2022) 24(6), 2669–2700. doi:10.1111/1462-2920.16045), the existence of a "non-enzymatic pathways of demethylation (Bentley and Chasteen, 2004)" is highly unlikely as is "DMS production by the biological reduction and oxidation of DMSP".

We modified the text according to the very recent review of Shaw et al. (2022):

"In the surface ocean, DMS is produced from the degradation of dimethylsulfoniopropionate (DMSP), which may be produced by marine macroalgae, phytoplankton or bacteria (Bentley and Chasteen, 2004; Kloster et al., 2006; Novak and Bertram, 2020). To form DMS, the DMSP undergoes reactions catalysed by DMSP lyase, (Taylor and Visscher, 1996; Steinke et al., 1996; see Shaw et al. 2022 for a review). **Other DMS production pathways have been suggested to occur from biological catabolism of DMSP and also abiotically by light-dependent reactions (McNabb and Tortell, 2021 and references therein), but the contribution of these pathways to marine DMS production is still an open question (Shaw et al. 2022). These multiple chemical and biological production pathways make numerical prediction of DMS fluxes complex.**"

- I acknowledge that the metabolism and catabolism of DMSP is complex as shown below but, unfortunately, the present manuscript does not make any advance in this field. The importance of nanophytoplankton in DMSP synthesis has been established 20 years ago.

The purpose of our measurements and paper is not to understand the mechanisms of DMSP synthesis. Although the importance of nanophytoplankton in DMSP production is established, our novel experimental system enabled us to quantify the relationship between nanophytoplankton and DMS fluxes at low wind speed, which can not be achieved in the laboratory and is difficult in the ambient environment, as stated in the introduction:

**"A major limitation when relating trace gases to sea-surface biota arises due to atmospheric mixing and transport, as the atmospheric emissions of DMS, MeSH and other trace gases in ambient air are spatially dislocated from their marine source (Bell et al, 2015). Here, we describe the use of a novel experimental set-up using Air-Sea Interface Tanks (ASIT) to measure DMS and MeSH fluxes in three different seawater types collected during a voyage in the South-West Pacific. While not representative of open ocean conditions in terms of wind-derived processes, the ASIT experiments employed in the Sea2Cloud voyage allowed direct observation of DMS and MeSH emissions and their respective contribution to total sulphur emissions, as well as their relationship with the biogeochemistry of the distinct seawater types. This experimental set-up also provided**

**the capacity to control some variables. As surface ozone has increased in clean Southern Hemisphere air over the last 30 years and is projected to continue to rise (Cooper et al. 2020), we evaluated the impact of ozone-mediated oxidative stress on these fluxes. In addition, the relationship between fluxes and seawater concentrations with physical, chemical and biological seawater properties were explored to develop DMS flux parameterizations for potential application in modelling efforts to constrain the spatial and temporal distribution of DMS fluxes in this region.**"

Although the importance of nanophytoplankton is established, we present the first parameterization, validated using natural phytoplankton communities, for modelling of DMS and MeSH fluxes. Furthermore, this study and other recent studies (e.g. Lawson et al 2020, Kilgour et al 2022) highlight the potentially significant contribution of methanethiol, a hitherto under-reported species, to the gas phase sulphur budget with possible implications for secondary marine aerosol formation.

- I agree with Dr. Dong's comment. The sea-air fluxes of DMS and CH3SH measured in the ASIT tanks by no means can be compared with literature values reported in Table 3, because the turbulence very likely is considerably lower in the tanks than in the reality.

We highlight in the text that the ASITs are a low turbulence environment that is not representative of the full range of ocean conditions; indeed the absence of wind and inclusion of baffles meant that turbulence was low and constant across the experiments, which enabled us to derive the relationship between flux and biology. Using the headspace-water gradient we were able to calculate a "windspeed" for the ASITs, which was very low relative to windspeed over the ocean. Consequently, we now include reported DMS flux measurements for the South Pacific Ocean only for regional relevance and do not compare these directly to our ASIT-derived fluxes. We do include all reported MeSH fluxes as there are very few available.

We now further acknowledge the low turbulence specific conditions under which fluxes are reported in Section 3.4:

"**Here we stress that these values are appropriate under low wind speeds (see section 2.6) and increased turbulence may have both direct and indirect effects on the biology-flux relationship.**"

We have section 3.6 "Representativeness of the ASITs DMS and MeSH fluxes" rewritten:

"**DMS fluxes measured in the ASITs, and also estimated from ambient concentrations (using the two-layers method and nocturnal boundary layer method described sections 2.6 and 2.7) are reported in Table 3, along with literature values. Using the COARE model and the Sea2Cloud seawater DMS data, Saint-Macary et al. (2023) estimated mean fluxes of 4.44 ± 1.05 ng m-2 s-1 for sub-Tropical front waters, 2.79 ± 0.83 ng m-2 s-1 for sub-Antarctic waters, and 1.83 ± 0.06 ng m-2 s-1 for sub-Tropical waters, which are approximately four times higher than those obtained from the ASITs. This is expected as flux is dependent on wind speed, which as noted was limited in the ASITs. In Hulswar et al., (2022), the authors determine DMS fluxes of around 5-10 µmol m-2 d-1 for the South-West Pacific region, which corresponds to the higher end of the range reported by Saint-Macary et al. (2023). DMS fluxes determined during the Sea2Cloud using the nocturnal boundary layer accumulation method were consistent with those estimated using the same method and in the same region during the SOAP voyages (Lawson et al., 2020). Both nocturnal boundary layer accumulation flux estimates are 1-2 orders of magnitude greater than the measured fluxes in the ASIT experiments. This high discrepancy reflects the low turbulence**

**conditions, and also other methodological assumptions in both approaches. For example, nocturnal boundary layer accumulation flux estimates may be biassed by long-range transport and vertical dilution which decouple ambient air from the underlying seawater. In addition to these physical factors, seasonal differences in phytoplankton community composition will also result in variability in fluxes."**

MeSH flux estimates are rare so we present all published estimates, but recognise that, as windspeed and turbulence are also important factors, the ASIT flux estimates will be lower than other estimates. We added the following sentence:

**"We acknowledge that, as windspeed and turbulence are important factors, the ASIT flux estimates will be lower and not representative of the open ocean. Although the fluxes cannot be directly compared the relative emissions of DMS and MeSH can."**

It is also worth noting that while a large variability in flux estimates are reported via different methods some consistent relationships are observed among studies notably the ratio of MeSH/DMS fluxes (range 0.03 - 0.24) which also holds for the ratio in ambient marine air (this study 0.09) (see Sect 3.6).

- By no means can this manuscript be entitled "Air-Sea fluxes of dimethyl sulphide and methanethiol in the South-West Pacific",

The ASITS do not give us a "air-sea flux" in the traditional but we do measure a flux or emission. We modified the title of our paper according to this comment: "Relating dimethyl sulphide and methanethiol fluxes to surface biota in the South-West Pacific using shipboard mesocosms".

- and the following sentences "Air-sea fluxes of dimethyl sulphide (DMS) and methanethiol (MeSH) from surface seawater in the remote Southern Pacific Ocean were measured in three Air-Sea Interface Tank (ASIT) experiments during the Sea2Cloud voyage in March 2020. The measured fluxes of $0.78 \pm 0.44$ ng m-2 s-1 and $0.05 \pm 0.03$ ng m-2 s-1 for DMS and MeSH, respectively, varied between experiments reflecting the different water mass types investigated, with lowest fluxes with subtropical water and highest with biologically-active water with sub-Tropical water and highest from the sub-Tropical Front" be put forward in the abstract.

The abstract now reads "**Dimethyl sulphide (DMS) and methanethiol (MeSH) emissions from South Pacific surface seawater were determined in three shipborne Air-Sea Interface Tank (ASIT) experiments during the Sea2Cloud voyage in March 2020. The measured fluxes varied between experiments reflecting the different water mass types sampled, with lowest fluxes observed from sub-Tropical (FDMS = 0.17 FMeSH = 0.04 ng m-2 s-1) and sub-Antarctic waters (FDMS = 0.51 FMeSH = 0.08 ng m-2 s-1) and highest fluxes with sub-Tropical frontal waters (FDMS = 1.44 FMeSH = 0.0.17 ng m-2 s-1).**".

- Another major flaw is the role attributed to ozone, i.e. "The experiments also determined the influence of elevated ozone, with one ASIT headspace amended with 10 ppbv ozone while the other provided an unamended control. Elevated ozone resulted in a decrease in DMS flux, corresponding to decreased conversion of dimethylsulfoniopropionate (DMSP) to DMS in the seawater". Concentrations of DMSP and DMS in seawater are displayed in Fig. S.5 and Fig. S.2, respectively (reproduced below). Although the poor quality of both figures makes the comparison difficult to be established, it seems that it is in ASIT-control not in ASIT-O3 that a major change (an increase in this case) in the DMS-to-DMSP ratio took place in EXP-A by Mar 22. How can one trust that

ozone is an inhibitor of the conversion of DMSP to DMS in seawater when the demonstration relies on a single observation? This process should have been investigated in the laboratory under more controlled conditions and in replicates.

As we state in the original text, the inhibitory effect of ozone on DMSP conversion to DMS is only a hypothesis. We now further acknowledge the limitations in our study and identify the need to further investigate this:
"**The nature of the ozone interaction is unclear, although it can be speculated that it influenced the release of intracellular DMSP possibly by suppressing bacterial/viral lysis of phytoplankton cells, and so limiting conversion to DMS. The ozone concentrations tested were within the typical range in the ambient atmosphere in this region, pointing to a potentially important metabolic influence of ozone. However, recognising the limited dataset and also the absence of replication in the ASIT experiments further work is required to confirm potential inhibition by ozone.**"

- This manuscript should be rejected in its present form.

We hope we took into account all issues raised by the reviewer.

---

## Author Comment (AC2)

**Answers to comments on "Relating dimethyl sulphide and methanethiol fluxes to surface biota in the South-West Pacific using shipboard mesocosms" article submitted in *ACP*.**

Manon Rocco[1,2*], Erin Dunne[3], Alexia Saint-Macary[4,7], Maija Peltola[1], Theresa Barthelmeß[5], Neill Barr[4], Karl Safi[4A], Andrew Marriner[4], Stacy Deppeler[4], James Harnwell[3], Anja Engel[5], Aurélie Colomb[1], Alfonso Saiz-Lopez[6], Mike Harvey[4,†], Cliff S. Law[4,7], and Karine Sellegri[1]

[1] Université Clermont Auvergne, CNRS, Laboratoire de Météorologie Physique (LaMP), 63000, Clermont-Ferrand, France
[2] Now at Instituto de Astronomia, Geofísica e Ciências Atmosféricas/IAG - Universidade de São Paulo - Laboratório de Estudos Ambientais/IPA, São Paulo, Brazil
[3] CSIRO Environment, Aspendale, Australia
[4] National Institute of Water and Atmospheric Research, Wellington, New Zealand
[4A] National Institute of Water and Atmospheric Research, Hamilton, New Zealand
[5] GEOMAR Helmholtz Centre for Ocean Research Kiel, Kiel, Germany
[6] National Center for Atmospheric Research, NCAR, Boulder, USA
[7] Department of Marine Science, University of Otago, New Zealand
[†] Deceased

*Correspondence to*: Karine Sellegri (karine.sellegri@uca.fr); Manon Rocco (rocco.manon@gmail.com)

**#Editor comments:**

After careful reading of the manuscript, I agree with the reviewer that the manuscript requires drastic improvement in order to meet the quality for publication in ACP.

- The authors need to properly cite earlier works both with regard to the understanding of the processes of DMS production in the sea water as well as to the DMS observations in the marine environment as already suggested by the reviewer.

The updated version of the Lana climatology is now mentioned in the introduction. This version is very recent (2022) and therefore was not available at the start of writing the present paper, which explains why it was missing. We now include it. However, we contest the reviewer's assessment that our introduction makes insufficient use of DMS state of the art . We use 32 different references, among which many are from the last 5 years (Novak and Bertram, 2020; McNabb and Tortell, 2021; Lawson et al., 2020; Kilgour et al., 2021; Wang et al., 2020;Galí et al., 2018; Bell et al., 2021).

- They also need to clearly demonstrate and spell out (also in the abstract) the originality and the added value of the work compared to earlier studies.

We provide the first quantitative relationships between emissions of DMS, and also MeSH, with nanophytoplankton abundance, derived across a range of oceanic water types.

We now specify this  in the abstract:

"Dimethyl sulphide (DMS) and methanethiol (MeSH) emissions from South Pacific surface seawater were determined in three **shipborne**."

and in the conclusion:

"We found that fluxes measured in the ASITs could be accurately predicted from seawater concentrations assuming a very low wind speed. The experimental set-up in our study provided a new approach to relating DMS and MeSH fluxes to the biogeochemical properties of surface seawater. Previous empirical relationships linking DMS fluxes and seawater biology have used Chl-a derived from satellite retrievals, which has inherent spatial resolution and biological limitations."

- In addition, they have to thoroughly discuss the uncertainties introduced by the contamination and losses inside the ASITs.

Regarding contamination, our fluxes are calculated as a difference of concentration between the headspace and flushing air, so any contamination coming from the flushing air would be subtracted (as a "blank"). Moreover, here we focus on DMS and MeSH, which do not suffer from contamination issues (from the system itself).

Furthermore, in a previous report from this study (Rocco et al., 2021) potential contamination from the ASITs and sampling system were investigated for aromatic VOCs which are commonly associated with man-made materials. The aromatic VOCs concentrations in 3 seawater samples collected from the ASITs were close to those measured in samples collected from open ocean water, indicating contamination from ASITs materials was not a significant artefact.

- For instance, they mention that the O3 levels inside the ASIT are lower than the ambient levels in the control experiment due to losses on the walls and the pipeline. This means that the control experiment is not representative of the ambient atmosphere – so the derived fluxes are not representative either.

A discussion was added section 2.1. on ozone:

"The ASIT-O3 ozone concentrations (14.5 ± 2.9 ppbv) were closer to the ambient air ozone concentrations (14.6 ± 1.8 ppbv) than the ASIT-control levels (6.7 ± 1.5 ppbv) due to potential wall losses of ozone in the tanks and sampling lines, but also potential increased reactivity in the ASITs headspaces compared to ambient air (as biogenic emission fluxes under low turbulence tend to be more concentrated than in ambient air). **Consequently, the ASITs experiments could only compare sub-ambient O3 with ambient and so, by combining data from both ASITs, measured fluxes were obtained under the typical ozone range observed over this oceanic region and during this period of the year (Bourgeois et al. 2020).**"

- In addition, if O3 is lost in the system why not having also artifacts for the other studied species?

We corrected the statement that lower ozone concentrations being lower in the ASITs are attributed to losses: "The ASIT-O3 ozone concentrations (14.5 ± 2.9 ppbv) were closer to the ambient air ozone concentrations (14.6 ± 1.8 ppbv) than the ASIT-control levels (6.7 ± 1.5 ppbv) due to **potential** wall losses of ozone in the tanks and sampling lines, **but also increased reactivity in the ASITs headspaces compared to ambient air (as biogenic emission fluxes under low turbulence enclosures tend to be more concentrated than in ambient air).**"

We also further discuss the potential losses of other species, such as DMS.

"The estimated headspace concentration of DMS calculated from the Henry's law equation (Eq. 1) showed an excellent correlation with DMS measured in the headspace (DMShs) ($R^2 = 0.94$, slope $=1.66$, intercept $= -0.75$), indicating equilibrium conditions were established in the ASITs (Figure 6), and so wall and chemical losses were limited under the flow-through experimental conditions with headspace concentrations reflecting changes in dissolved DMS concentrations in the underlying seawater."

- However, I cannot find an evaluation of how the errors induced by this artifact propagate to the main findings of the study. I would also like to see a more critical presentation of the results of the ASIT experiments since only 3 pairs of them have been performed. Even though these experiments are logistically heavy to be performed, it is difficult to draw a firm conclusion from a such small number of experimental results and thus the way the results are presented has to be appropriate.

Each experiment lasts several days, and for our study of relationships between chemical concentrations (or fluxes) and seawater biogeochemical variables, we merged both ASIT's data which doubles the number of points. We do provide a statistical analysis of the robustness of our relationships (p values) provided Table 3.

Regarding the impact of ozone, we clearly acknowledge the small number of data points, and lack of duplicates: "**However, recognising the limited dataset and also the absence of replication in the ASIT experiments further work is required to confirm potential inhibition by ozone.**".

- Furthermore, the authors erroneously mention a lifetime of DMS for its reaction with O3 at 15 days with reference to Vrekoussis et al., 2004, who do not mention any relevant to that result since that study investigated OH and NO3 radicals atmospheric levels. To my knowledge there is only a very low upper limit rate for reactivity against O3. For a thorough review of DMS chemistry – although a bit old now -but very comprehensive the authors are directed to the Barnes et al Chemical Reviews paper Chem. Rev. 106, 940-975, 2006.and recent updates by Veres et al. 2021 https://doi.org/10.1073/pnas.1919344117

This is right. We now refer to Burkholder et al. (2015) following Fung et al. (2022). The reactivity of DMS with ozone leads to a chemical lifetime of more than 12 years.

| Gas-phase reactions | $k_{298}$ (cm$^3$ molec.$^{-1}$ s$^{-1}$) | $-E_a/R$ (K) | References |
|---|---|---|---|
| $DMS + OH \rightarrow 0.6SO_2 + 0.4DMSO + CH_3O_2$ | See footnote* | – | Burkholder et al. (2015), Pham et al. (1995) |
| $DMS + NO_3 \rightarrow SO_2 + HNO_3 + CH_3O_2 + CH_2O$ | $1.13 \times 10^{-12}$ | 530 | Burkholder et al. (2015) |
| $DMS + BrO \rightarrow DMSO + Br$ | $3.39 \times 10^{-13}$ | 950 | Burkholder et al. (2015) |
| $DMS + O_3 \rightarrow SO_2$ | $1.00 \times 10^{-19}$ | 0 | Burkholder et al. (2015) |
| $DMS + Cl \rightarrow 0.5SO_2 + 0.5DMSO + 0.5HCl + 0.5ClO$ | $3.40 \times 10^{-10}$ | 0 | Burkholder et al. (2015) |
| $DMSO + OH \rightarrow 0.95MSIA + 0.05SO_2$ | $8.94 \times 10^{-11}$ | 800 | Burkholder et al. (2015) |
| $MSIA + OH \rightarrow 0.9SO_2 + 0.1MSA$ | $9.00 \times 10^{-11}$ | 0 | Burkholder et al. (2015) |
| $MSIA + O_3 \rightarrow MSA$ | $2.00 \times 10^{-18}$ | 0 | Lucas (2002) |

The Barnes et al. review only mentions the DMS reactivity towards OH, NO3 and halogenes; it does not contain any information regarding reactivity with ozone. We feel it would be out of scope to include

a review of the DMS reactivity with OH, as in our case most of our DMS fluxes were measured during nighttime. As the Veres et al. study investigates a new pathway which is also initiated by OH, it is also irrelevant here.

- There are also inconsistencies between the numbers provided in the text and the figures as in Figures 4 and 5 pointed out by the reviewer. Although the color scale in Figure 4 does not allow to see the exact value of the concentrations, the high levels of DMS provided in the manuscript, for instance the 1285 ppt of DMS in line 269. In addition, alteration between ppt and ppb for DMS concentration in the text and the figures is rather confusing for the reader.

The value of 1285 ppt has been updated: "**Mixing ratios of DMS ranged from below the detection limit (< 78 pptv) to 753 pptv with a voyage average of 171 ± 118 pptv**". The unit was changed following the reviewer comment and all concentrations are now given in ppt.

- Furthermore, key references. like Sellegri et al 2923 and Rocco et al. 2021, are provided in the text and are missing from the ref list. The first one, I imagine, should be the Sea2Cloud description published in https://journals.ametsoc.org/view/journals/bams/104/5/BAMS-D-21-0063.1.xml and the second should be the paper in Nature Communications E&E https://www.nature.com/articles/s43247-021-00253-0 where ASIT results are also presented for organics.

Yes, this is now corrected.

The manuscript will also benefit from a careful re-reading and English correction.

- Overall, I consider this manuscript requires a very careful examination of all its statements, the statements need to be supported by observations and statistical analysis (including p-values for all correlations),

Again, p values are provided and therefore we believe our conclusions are supported by a careful statistical analysis.

- and needs to demonstrate its added value compared to earlier studies from different or even the same campaign.

We hope this is now much clearer.

- In addition, the content should reflect on the title. As is now, focused on the ASIT experiments, the title in the supplementary material seems closer to the content of the manuscript than the original title.

The title now reflects the content of the paper.

Bourgeois, I., Peischl, J., Thompson, C. R., Aikin, K. C., Campos, T., Clark, H., Commane, R., Daube, B., Diskin, G. W., Elkins, J. W., Gao, R.-S., Gaudel, A., Hintsa, E. J., Johnson, B. J., Kivi, R., McKain, K., Moore, F. L., Parrish, D. D., Querel, R., Ray, E., Sánchez, R., Sweeney, C., Tarasick, D. W., Thompson, A. M., Thouret, V., Witte, J. C., Wofsy, S. C., and Ryerson, T. B.: Global-scale distribution of ozone in the remote troposphere from the ATom and HIPPO airborne field missions, Atmos. Chem. Phys., 20, 10611–10635, https://doi.org/10.5194/acp-20-10611-2020, 2020.

Burkholder, J. B., Sander, S. P., Abbatt, J. P. D., Barker, J. R., Huie, R. E., Kolb, C. E., Kurylo, M. J., Orkin, V. L., Wilmouth, D. M., and Wine, P. H.: Chemical kinetics and photochemical data for use in atmospheric studies: evaluation number 18, Jet Propulsion Laboratory, National Aeronautics and Space Administration, Pasadena, CA, USA, http://hdl.handle.net/2014/45510 (last access: 30 November 2021), 2015.

Fung, K. M., Heald, C. L., Kroll, J. H., Wang, S., Jo, D. S., Gettelman, A., Lu, Z., Liu, X., Zaveri, R. A., Apel, E. C., Blake, D. R., Jimenez, J.-L., Campuzano-Jost, P., Veres, P. R., Bates, T. S., Shilling, J. E., and Zawadowicz, M.: Exploring dimethyl sulfide (DMS) oxidation and implications for global aerosol radiative forcing, Atmos. Chem. Phys., 22, 1549–1573, https://doi.org/10.5194/acp-22-1549-2022, 2022.